



# Advancing $N_2O$ flux chamber measurement techniques in nutrient-poor ecosystems

Nathalie Ylenia Triches [1,2], Jan Engel [1], Abdullah Bolek [1], Timo Vesala [3], Maija E. Marushchak [4], Anna-Maria Virkkala [5], Martin Heimann [1], and Mathias Göckede [1]

[1]Department of Biogeochemical Signals, Max Planck Institute for Biogeochemistry, Jena, Germany
[2]Institute for Atmospheric and Earth System Research/Forest Sciences, Faculty of Agriculture and Forestry, University of Helsinki, Helsinki, Finland
[3]Institute for Atmospheric and Earth System Research/Physics, Faculty of Science, University of Helsinki, Helsinki, Finland
[4]Department of Environmental and Biological Sciences, Faculty of Science, Forestry and Technology, University of Eastern Finland, Kuopio, Finland
[5]Woodwell Climate Research Center, Falmouth, USA

**Correspondence:** Nathalie Ylenia Triches (ntriches@bgc-jena.mpg.de)

**Abstract.** Nitrous oxide ($N_2O$) is the third most important greenhouse gas whose atmospheric mole fraction has risen from 273 ppb to 336 ppb since 1800, mainly due to agricultural activities. However, nutrient-poor natural soils, including those in the (sub-) Arctic, also emit and consume $N_2O$. These have not been investigated thoroughly, partly because of methodological limitations in reliably detecting low fluxes. These limitations were, to a large extent, driven by the available instrumentation:

lacking portable gas analysers for $N_2O$ with appropriate accuracy, researchers relied on manual air sampling from closed flux chambers, with subsequent analysis using gas chromatographs (GC) in a laboratory. In this study, we use a fast-responding, portable gas analyser (PGA; Aeris $N_2O/CO_2$) combined with a custom manual chamber system, incorporating both transparent (light) and opaque (dark) measurements, for its suitability to measure low $N_2O$ fluxes from a nutrient-poor, sub-Arctic peatland. We assess the performance of the analyser under low-flux conditions, evaluate the effects of chamber closure time, and compare

linear and non-linear models for quantifying concentration gradients. Moreover, we compare flux rates based on high-frequency *in situ* observations against an approach that randomly draws discrete samples from the full time series, mimicking a GC-based approach. Our results show that with our PGA, we can successfully detect and calculate low $N_2O$ flux rates, with a mean of $12.9 \pm 28.4 \, \text{nmol m}^{-2}\text{h}^{-1}$ under light conditions and $-46.1 \pm 38.2 \, \text{nmol m}^{-2}\text{h}^{-1}$ under dark conditions, depending on chamber closure time. The majority of fluxes (88% for light and 74% for dark measurements) exceeded the minimum detectable flux

(MDF), which was $14.5 \pm 1.05 \, \text{nmol m}^{-2}\text{h}^{-1}$ for light and $14.7 \pm 1.08 \, \text{nmol m}^{-2}\text{h}^{-1}$ for dark measurements. Our comparison of chamber closure times (3–10 min) showed that 3 minutes may be insufficient for capturing low $N_2O$ fluxes during light measurements, while closure times of 4–10 minutes provide more reliable results. For dark measurements, where $N_2O$ uptake was highest with short closure times, we recommend a chamber closure time of 3–5 minutes, unless data are limited, in which case longer times may help capture fluxes above the MDF. In our study, all $N_2O$ fluxes were calculated using the non-linear

model or matched the linear model when data showed a linear distribution. Compared to the PGA-based flux calculations, GC-simulations underestimated $N_2O$ fluxes when using 3-6 samples. Therefore, we suggest that fast-responding analysers may be better suited to measure low $N_2O$ fluxes and improve the understanding of diverse $N_2O$ flux dynamics.



## 1 Introduction

Nitrous oxide ($N_2O$) is the third most important greenhouse gas (GHG) with a global warming potential almost 300 times
stronger than carbon dioxide ($CO_2$) over a period of 100 years (Intergovernmental Panel On Climate Change (Ipcc), 2023). It
stays in the atmosphere for around 110 years and acts as an ozone-depleting compound in the stratosphere, causing direct harm
to humans (Intergovernmental Panel On Climate Change (Ipcc), 2023). The atmospheric mole fractions of $N_2O$ have increased
from 273 ppb to 336 ppb since 1800 (Thoning et al., 2022). As most of this increase can be attributed to human activities,
research has focused on $N_2O$ emissions from managed, agricultural soils (De Klein et al., 2020) that hold the potential for
high $N_2O$ emissions. This is because the input of nitrogen fertilisers in managed soils increases the readily available mineral N
needed for plant growth and thus increases harvest, but, simultaneously, can also result in higher $N_2O$ emissions (Myhre et al.,
2013).

Until about 15 years ago, only few studies investigated $N_2O$ fluxes in the (sub) Arctic, where soils often have a very low
availability of reactive N (Virkkala et al., 2024) and thus are not expected to emit or take up amounts of $N_2O$ relevant for the
global climate (Voigt et al., 2020; Christensen et al., 1999; Grogan et al., 2004). Since 2009, multiple studies have reported $N_2O$
emissions similar to agricultural soils from organic-rich ecosystems in the Arctic (Repo et al., 2009; Marushchak et al., 2011;
Elberling et al., 2010), shifting the focus to only selected, high-nutrient areas within the (sub) Arctic. Nevertheless, reporting
near-zero $N_2O$ fluxes is crucial to avoid overestimating emissions caused by biased site selection favoring high-emitting areas
(Voigt et al., 2020).

In most studies, $N_2O$ concentrations were sampled repeatedly with a syringe from the head space of a closed flux chamber
and measured with a gas chromatograph (GC) in the laboratory (Hensen et al., 2013; Denmead, 2008; Pavelka et al., 2018).
With this approach, typically between 4 and 9 discrete air samples are taken to measure the trend in $N_2O$ mixing ratios
during chamber closure time and calculate the fluxes. The sensitivity of GCs varies, but with only 4-9 samples drawn from
a fluctuating time series that may not necessarily display a linear trend, differences in low concentrations are hard to capture
and highly dependent on single data points (Hübschmann, 2015). Additionally, in previous studies, opaque chambers have
been mostly used because temperature inside the chamber would increase less above the ambient temperature compared to
transparent chambers (Clough et al., 2020). As a result, there are only few studies investigating fluxes under different light
conditions (Stewart et al., 2012). Since this was the only available method for *in situ* $N_2O$ measurements in the field, our
knowledge on (sub) Arctic $N_2O$ fluxes is rather limited and makes it challenging to establish accurate baseline estimates,
which are essential for detecting changes in fluxes.

Recent advances in laser spectroscopy led to novel, portable (< 15 kg) and fast-responding (1 Hz, *i.e.,* sampling every second)
GHG analysers, offering new possibilities to measure low $N_2O$ concentrations in nutrient-poor ecosystems (Subke et al., 2021).
These portable gas analysers (PGA) allow near-continuous monitoring of concentration changes, providing higher precision
and lower detection limits than GC-based methods (Hensen et al., 2013). While differences in flux estimates between PGA and
GC have been well-documented for $CH_4$ and $CO_2$, few studies have focused on $N_2O$ (Christiansen et al., 2015; Brümmer et al.,
2017). At the same time, many of the reported $N_2O$ fluxes were found to be below the detection limit, making it challenging to



assess the magnitudes and possible uptake of these fluxes. It is crucial to know at which accuracy $N_2O$ fluxes can be measured, and how to best measure them.

The availability of portable gas analysers for *in situ* $N_2O$ flux measurements raises new methodological questions. First, in

contrast to $CH_4$ and $CO_2$, where a chamber closure time of 3 min or less is well-accepted, it remains unclear how long chambers need to be closed for reliable $N_2O$ flux measurements in nutrient-poor ecosystems. This is due to the low concentrations of $N_2O$, which need more time than $CH_4$ and $CO_2$ to accumulate in the chamber head space in order to reach a sufficiently high change in $N_2O$ concentration to detect a significant trend over the instrument noise and to reliably calculate the fluxes (Fiedler et al., 2022). Few studies have investigated the chamber closure time with portable $N_2O$ analysers, and the reported

recommendations range between 3 and 10 min (Fiedler et al., 2022; Brümmer et al., 2017). Second, there is a long-ongoing discussion within the chamber community on whether to use linear (LM) or non-linear (HM) models to calculate flux rates (Kutzbach et al., 2007), which has not been investigated for low $N_2O$ fluxes. The critique on the linear models is that they underestimate the flux rates due to the assumption that GHG concentrations keep increasing within the chamber head space (Fiedler et al., 2022). However, it is clear from the theory of molecular diffusion that the rate of concentration change within

the chamber declines over time (Hutchinson and Mosier, 1981; Kutzbach et al., 2007; Kroon et al., 2008), which then leads to the underestimation of fluxes when using linear models (Hüppi et al., 2018). As a result of that, there have been great efforts to implement non-linear flux calculations as alternatives for LM, for example, through software packages (Pedersen et al., 2010; Hüppi et al., 2018). Nevertheless, the use of non-linear flux calculations is still not a standard within the chamber community, most likely due to its complexity.

The main aim of this paper is to provide a first, extensive data set of $N_2O$ fluxes measured with our measurement system consisting of a portable $N_2O$ analyser and both transparent and dark flux chambers to quantify low $N_2O$ in a nutrient-poor ecosystem. We tested the instrument performance of our PGA in the laboratory and in the field, with our data set covering various land cover types from a thawing permafrost peatland in sub-Arctic Sweden. We compare $N_2O$ flux rates across different chamber closure times (3 min- 10 min) and evaluate differences between linear and non-linear calculation methods. Moreover,

we compare flux rates based on high-frequency *in situ* observations against an approach that randomly draws discrete samples from the full time series, mimicking a GC-based approach. Finally, we aim to provide guidance on measuring $N_2O$ fluxes in nutrient-poor ecosystems, such as the Arctic. Ideally, this will encourage researchers to measure low fluxes in sub-Arctic regions, get a better process understanding of $N_2O$ fluxes, and determine how the N cycle in nutrient-poor ecosystems will react to global warming.

## 2   Methods

To facilitate the reader's understanding, we use the terminology proposed by Fiedler et al. (2022), with location describing the area where sampling occurs ("Stordalen mire"), site describing a vegetation unit within the location ("palsa lichen", "palsa moss", "bog", "fen"), and chamber base position (*i.e.*, plot) for the exact spot where $N_2O$ was measured. With "chamber closure time", we specify the time frame a chamber was closed onto the soil; one of these periods is then called "measurement period".





## 2.1 Study location and sampling sites


All data were collected at the Stordalen mire, a complex palsa mire underlain by sporadic permafrost located in subarctic Sweden (68° 200 N, 19° 300 E), 10 km east of Abisko (Ábeskovvu in Northern Sámi language). Permafrost has been rapidly thawing at this location over the last decades, and only remains in the dry uplifted areas on the peatland (palsas) (Sjögersten et al., 2023). For our study, we randomly selected 24 chamber base positions in three transects on a dry-to-wet thawing gradient
from palsa to bog to fen, with 6 replicates for each land cover type: palsa lichen, palsa moss, bog, and fen (Fig. 1). Transects 1 and 2 each contain 6 chamber base positions and are located in the northern center of the mire, within the footprint of an Integrated Carbon Observation System (ICOS, SE-Sto) eddy covariance tower which has been operating since 2014 (Lundin et al., 2024), Fig. 1). Transect 3 lies in the most north-eastern part of the palsa.

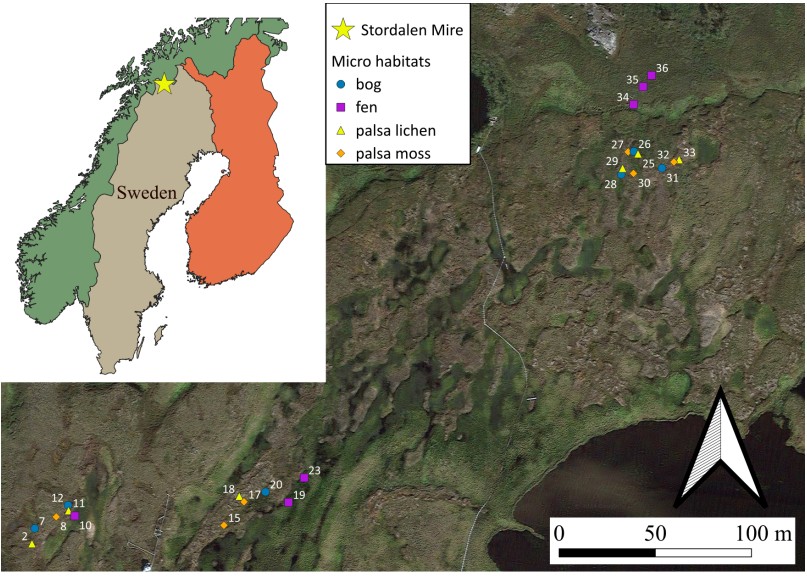

**Figure 1.** Three transects with chamber base positions in Stordalen (overlaid on satellite image from © Google Maps). The location of the Stordalen Mire (the shape files of each country can be found at https://simplemaps.com, last acess: 17/09/2024) is marked with a star. Here, each micro habitats are represented with different colors and symbols for clarity.

Vegetation on the palsa is mainly dominated by lichen (*Cladonia spp.*), shrubs (*Empetrum hermaphroditum*, *Betula nana*,
*Vaccinium uliginosum*, *Vaccinium vitis-idaea*, *Rubus chamaemorus*) and some mosses (*Dicranum elongatum*, *Sphagnum fuscum*). Both bogs and fens contain peat-forming mosses (*Sphagnum balticum*, *Sphagnum lindbergii*, *Sphagnum riparium*), with the dominant vascular plants on fens being cotton grass (*Eriophorum vaginatum*, *Eriophorum angustifolium*) and in bogs sedges (*Carex rotundata*, *Carex rostrata*). The soils in the area are classified as organic histosols or, if permafrost occurs within 2 m of cryoturbation activities, as cryosols (Siewert, 2018). Research at the Stordalen mire has been conducted for over a
century (Jonasson et al., 2012; Callaghan et al., 2013), and a vast amount of data on $CH_4$ and $CO_2$ fluxes has been published





(Łakomiec et al., 2021; Varner et al., 2022). The mean annual temperature at the Stordalen mire is -0.6°C and the annual precipitation 304 mm (Malmer et al., 2005).

The data presented in this study were collected during three separate campaigns covering different seasons: spring (between 23 - 30 May 2023), summer (20 - 27 July 2023), and autumn (3 - 22 September 2023). PVC collars with an inner diameter of 110   245.1 mm, a height of 150 mm, and a wall width of 4.9 mm were inserted into the soil on 29 August 2022. We inserted them as deeply as possible, between 100- 140 mm, to ensure a proper seal between the chamber head space and the atmosphere even during strong wind conditions, and in the palsa where the top peat was dry and highly porous. Between the collar and the chamber, a custom made sealing ring was placed to avoid ambient air entering the chamber during our measurements (Fig. 2, S2). The sealing ring has an inner and outer diameter 235 and 265 mm, respectively, a height of 30 mm and is build from 115   a metal ring wrapped in foam material (50 mm on each side). Tests confirmed that it sealed the chamber and the collar even under high wind conditions with up to 18 $ms^{-1}$ wind gusts.

## 2.2   Chamber and portable gas analyser (PGA) setup and protocol

For our measurements, we used a custom-built static, non-steady state, non-flow-through chamber (Livingston and Hutchinson, 1995) made from acrylic glass (Göli GmbH) with a height of 250 mm and a diameter of 250 mm (Fig. 2, S1). A fan (SUNON 120   Maglev, 80 mm x 80 mm x 25 mm, 2000 RPM) was installed inside the chamber to ensure well mixed conditions within the chamber during the measurements. Additionally, a relative humidity (RH) and temperature probe (EE08, E+E Elektronik, Germany) and a pressure sensor (61402V, RM Young) were installed for measuring essential parameters to calculate the fluxes. The chamber was equipped with quick-release connectors on top to connect the inlet and outlet tubing to the portable gas analysers. As complementary variables, we measured soil temperature at 15 cm (PT100 4-wire sensors, JUMO GmbH & Co. 125   KG) at each quadrant outside of the plot, soil moisture at 12 cm and 30 cm (CS655-DS and CS650-DS, Campbell Scientific), and photosynthetically active radiation (PAR) (PQS1, Kipp and Zonen). More detailed information on our chamber setup can be found in the supplementary information.




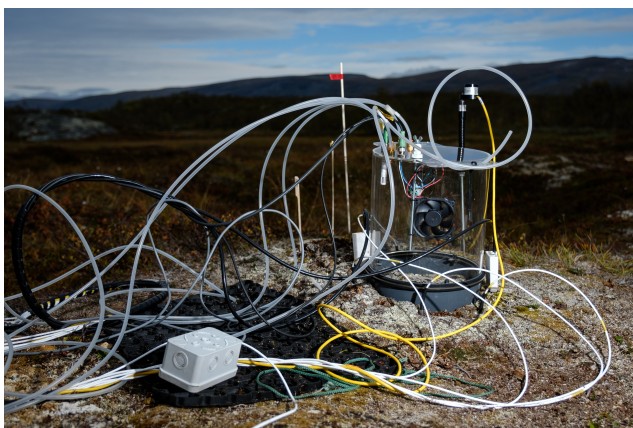

**Figure 2.** Chamber setup during measurement period, with soil moisture and soil temperature sensors installed in the soil, and all inlets connected. Photo: Fabio Cian

To measure $N_2O$ concentrations, we used the Aeris MIRA Ultra $N_2O/CO_2$ (from now onward: Aeris-$N_2O$) analyser (Aeris Technologies; sensitivity: 0.2 ppb/s for $CO_2$ and $N_2O$, frequency: 1 Hz). We performed several laboratory tests to assess the
signal stability (*i.e.*, drifts and stabilisation time), uncertainties, noise level, and water interference of the Aeris-$N_2O$. The analyser was left to sample ambient air for approximately 15 hrs to evaluate the signal stability (see section 3.1). Furthermore, we tested the impact of the humidity on the Aeris-$N_2O$ analyser using a portable dew point generator (LI-610, Licor USA). By adjusting the dew point temperature, we examined four different humidity levels: 28, 45, 60, and 83 %. A calibration gas tank with a known $N_2O$ concentration of 333.2 ppb was first connected to the dew point generator. The humidified gas was
then connected to the Aeris-$N_2O$ analyser inlet and each humidity level was sampled for about 20 minutes. Nevertheless, only a 10-minute window was used for further analysis due to relatively long time (about 10 mins) required for the humidity levels to stabilise (see Fig. A2).

In the field, we attached a custom made external battery box with two LiFePO$_4$ batteries (LiFePO$_4$ 12.8V 20Ah, Green Cell) to the Aeris-$N_2O$, which could be switched whilst the analyser was running. In this study, one LiFePO$_4$ 12.8V 20Ah battery
lasted for the whole day of field measurements (8h - max. 12h). A data logger (CR1000X, Campbell Scientific) was used to log all the sensor data including greenhouse gas concentrations which was placed inside a Pelican-case (Fig. S3 and S4). All GHGs and explanatory variables were logged with a frequency of 1 Hz. With this setup, all necessary information for the analyses was synchronised in a single data file, rather than many individual files from individual sensors and loggers.

Before we started a measurement period (*i. e.*, time when chamber is closed), we attached the tubes from the PGA to the
chamber, ventilated the chamber for at least one minute, and gently closed it onto the sealing ring. The default chamber closure time for all measurement periods was 10 minutes. For the dark measurements, a custom-made, reflective, light-impermeable tarp, isolating against temperature increases, was placed on top of the chamber.

All data were processed in R (version 4.3.3; R Core Team, 2024) and version controlled in GitLab (the scripts are publicly available from https://git.bgc-jena.mpg.de/ntriches/data-analysis/-/tags/2024-12-12-triches-amtsubmission-n2oadvances). A fil-





ter script was applied to pre-process and quality-control the raw data (*i.e.*, $N_2O$, $CO_2$, and $H_2O$ concentrations, chamber pressure, chamber temperature, chamber relative humidity, soil temperature, volumetric water content, and PAR), such as removing data points within a specific time-interval at the start of the measurement period to account for the time lag of gases moving through the tubes to reach the laser cell, and detecting erroneous data due to instrument failure, seen as flat lines, *i. e.*, periods of exact same concentrations (see SI 1.3). The filter script also included quality control of other parameters by, *e. g.*, remov-

ing implausible values (*e. g.,* -9999), replacing negative PAR values with 0, averaging soil temperature gained from the four sensors, and setting minimum and maximum values for all parameters (see SI). Furthermore, we used this script to simulate differences between chamber closure times and simulated gas chromatograph (GC) sampling, and the associated sensitivity analysis.

## 2.3 Flux calculations

In this study, fluxes were calculated using *all data points* from one measurement period. We removed 8 s in the start of the measurement period to account for the time delay until the concentration from the chamber reached the cell of the online laser analysers. An extra 7 s were removed for dark measurements, since we needed more time in the field to cover the chamber with the reflective tarp. To calculate the fluxes with both linear (LM) and non-linear (HM) methods in a reproducible way, we used the goFlux R package (v0.2.0, (Rheault et al., 2024)). We selected goFlux for several reasons: (i) it was specifically written

to process data gathered with portable analysers, (ii) it uses measured temperature and pressure inside the chamber for flux calculation, (iii) it corrects for the dissolved gases in the water vapour inside the chamber, and (iv) it calculates fluxes using both LM and HM flux calculation methods. It further allows to compare results gained from LM and HM models through different statistical methods, flags, *e. g.*, the minimal detectable flux (MDF, Eq. 4), and creates plots for visualisation. For the flux calculation in LM, goFlux applies the commonly used linear equation as follows in Eq. 1:

$$F(t) = \frac{dC(t)}{dt}\frac{V}{A} \tag{1}$$

where F(t) is the gas flux rate at a given location during the chamber closure time (t), $\frac{dC(t)}{dt}$ is the mass concentration change with time, $V$ is the volume of the chamber, and $A$ the area of the soil covered by the collar (Subke et al., 2021).

The HM model approach is based on the Hutchinson and Mosier (1981) approach as given in Eq. 2:

$$C(t) = \varphi + (C_0 - \varphi)e^{-\kappa t} \tag{2}$$

here, $\varphi$ is the assumed constant gas concentration of the source within the soil (Pedersen et al., 2010), $C_0$ is the gas concentration in the chamber at the moment of chamber closure, and $\kappa$ is the model parameter. To limit $\kappa$ with a maximum threshold $\kappa_{\max}$, Eq. 3 was adapted from Hüppi et al. (2018).

$$\kappa_{\max} = \frac{F(t)}{\text{MDF}\,t} \tag{3}$$





Here, MDF is used to specify whether the flux estimate was above or below the detection limit which is based on the instrument
precision ($\eta$) and can be calculated using Eq. 4.

$$\text{MDF} = \frac{\eta}{t}\,\theta \tag{4}$$

where, $\theta$ is a flux term that corrects for the water vapor inside the chamber and converts the flux unit to $\mu$mol m$^{-2}$ s$^{-1}$, which
was calculated as given in Eq. 5

$$\theta = \frac{(1 - C_{\mathrm{H_2O}})V\,P}{A\,R\,T} \tag{5}$$

where $C_{\mathrm{H_2O}}$ is the water vapour concentration in mol mol$^{-1}$, $P$ is the pressure inside the chamber in kPa, $R$ is the universal
gas constant in L kPa K$^{-1}$ mol$^{-1}$, and T is the air temperature inside the chamber in K.

In the goFlux package, the fluxes that are below the detection limit are flagged. Note that owing to this function, all our flux
estimates have their own MDF value. The package further implements the so called g-factor ($g_f$) (Hüppi et al., 2018) to restrict
large curvatures of the non-linear flux models as follows (Eq. 6):

$$g_f = \frac{HM_F}{LM_F} \tag{6}$$

Here, $HM_F$ and $LM_F$ are the flux values that are calculated by HM and LM models, respectively. In this study, we used a
$g_f$ of 4, meaning that the flux calculated by the HM model can be max. 4 times higher than the flux calculated by the LM to
avoid a large overestimation of fluxes (Eq. 6). We used this factor because it has been previously used (Leiber-Sauheitl et al.,
2014). We did not use the mean absolute error nor the root mean square error to define if the HM or LM model performed
better, since they were very similar amongst all measurement periods. We did also not use $R^2$ as a filter criteria since low and
non-linear fluxes inherently results in low $R^2$ values (Kutzbach et al., 2007).

### 2.4 Data processing and simulations

We used our openly available script to simulate differences between chamber closure times and gas chromatograph (GC)
sampling, and the associated sensitivity analysis. Firstly, we calculated all fluxes with the original chamber closure time of
10 minutes (prec = 0.2, g.limit = 4). To asses the impact of different chamber closure times on N$_2$O fluxes, we then removed
data points at the end of the original measurements, thus looking at the same measurement periods using the first 3 min, 4
min, 5 min, 6 min, 7 min, 8 min, and 9 min of the original data. We then explored variations in the flux rates between light
(transparent) and dark (opaque) measurements as affected by the chamber closure time, and defined the number of fluxes
above the minimal detectable flux according to the goFlux output files. While calculating our fluxes, we became aware of one
chamber base position acting as a hot spot, *i. e.*, showing much higher flux rates than the other chamber base positions. Since
we wanted to focus our analyses on low fluxes, we removed this hot spot from all analyses.





The simulations of GC measurements are based on drawing discrete sub-samples from the continuous *in situ* time series from the PGA, mimicking a sampling scheme where information on the increase of gas concentrations during chamber closure time is limited to a few snapshots in time. We simulated four scenarios: 3 GC samples (taken at 5 min, 7 min, and 10 min), 4 GC samples (3 min, 5 min, 7 min, and 10 min), 5 GC samples (1 min, 3 min, 5 min, 7 min, and 10 min), and 6 GC samples (1 min, 3 min, 5 min, 7 min, 8 min, and 10 min). For simulating the GC concentration, we picked the time stamp as defined above and took the average of 10 sec of $N_2O$ concentrations measured by the PGA, *i. e.*, 5 seconds before and after the defined time stamp, as single GC point measurement (Fig. A3). We then calculated the resulting fluxes with goFlux (prec = 1.9, g.limit = 4) using a precision of 1.9 ppb according to sensitivity tests on an instrument at our laboratory (Agilent Technologies, 7890 B GC System, mean $N_2O$ concentration 395.746 ppb with a SD of 1.875 ppb over ten repetitions), before we plotted the simulated versus original flux concentration measurements. To evaluate the performance of each sampling scheme, we compared the slopes, p-values, and $R^2$ values between the simulations and original data. To get an estimate on the uncertainties associated with this GC simulation, we conducted a sensitivity analysis, where we did 21 simulations with a randomised selection of the 4 min sampling time (4 samples at 3,5,7,9 min), allowing a window of 30 sec around each selected GC point.

## 3   Results and discussion

### 3.1   Laboratory tests with the Aeris-$N_2O$

From the 15-hour ambient air sampling, we observed that the water vapour mole fraction in the ambient air dropped from approximately 2500 ppm to about 800 ppm within the first 30 min. It continued to decrease progressively throughout the sampling period; however, after 5 hours, the water vapour mole fraction somewhat stabilised, with a mean $H_2O$ concentration of 476.9 ppm with a standard deviation of 18.7 ppm for the rest of the sampling. Note that the observed changes in water vapour might be partly due to the analysers warming up period. Therefore, we discarded the initial 5 hours of data and used the remaining data to assess the signal stability and noise characteristics of the Aeris-$N_2O$. The Aeris-$N_2O$ demonstrated a very stable signal with no apparent drift for about 10 hours of sampling period, having a low standard deviation of 0.324 ppb. Using Allan deviation plots (Allan, 1987), we evaluated the analyser's noise characteristics and found that the Aeris-$N_2O$ showed low instrument noise, approximately 0.16 ppb at 2-second averaging (see Fig. 3).



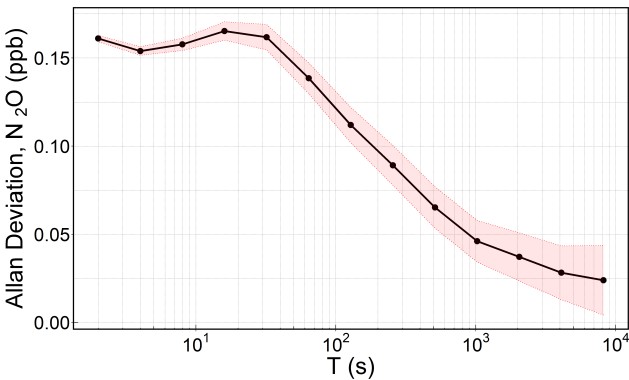

**Figure 3.** Computed Allan deviation plot based on 10 hours of continuous sampling, following a 5-hr spin up period during which the water vapor mole fraction was not stable. Here, T is the sampling time in log-scale and shaded region represents the 95% confidence interval.

Due to observed relatively high fluctuations under changing water vapor mole fraction, we tested the Aeris-$N_2O$ against different relative humidity (RH) conditions. Our tests with the four RH values (approx. 28%, 45%, 60% and 83%, respectively) showed that the water interference of the Aeris-$N_2O$ was very small; only slight differences were observed in the mean $N_2O$ concentrations for each humidity level (see Fig. 4), with mean $N_2O$ concentrations of 332.7 ppb, 332.6 ppb, 332.7 pbb, and

332.5 ppb for RH values of 28%, 45%, 60%, and 83%, respectively. Furthermore, we observed the same standard deviation of about 0.3 ppb for each humidity level. Overall, the conducted laboratory tests indicated that the Aeris-$N_2O$ was a suitable instrument for measuring low $N_2O$ fluxes, showing low noise and water interference, along with negligible signal drift after the laser warms up.

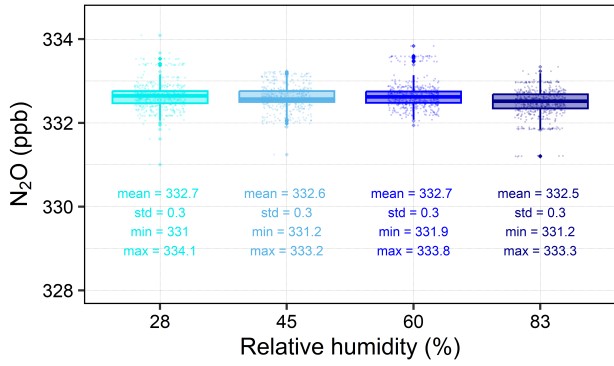

**Figure 4.** Measured $N_2O$ concentrations for different relative humidity (RH) values, with basic statistics of each RH values summarised under each box plots. Each humidity level was sampled for about 20-mins; however only a 10-min window was used for further calculations. The test used a standard gas with a mean of $333.16 \pm 0.16$ ppb as input. Jittered points are overlaid on the boxplots to visually separate overlapping data points, illustrating the distribution and density of the data.



## 3.2 Impact of chamber closure time on $N_2O$ fluxes

To report our flux rates, we use negative signs for an uptake of $N_2O$ into the soil, and positive signs for emissions. Our study site acted as a sink of $N_2O$, with a mean flux of -0.469 $\pm$ 1.25 $\mu$g $N_2$O-N m$^{-2}$ h$^{-1}$ during a chamber closure time of 10 min. Separating light and dark measurements, we get 0.361 $\pm$ 0.797 and -1.29 $\pm$ 1.07 $\mu$g $N_2$O-N m$^{-2}$ h$^{-1}$ during a chamber closure time of 10 min, respectively (Table 1). With up to 170 measurement periods for light and dark measurements, *i.e.*, 338 times we closed the chamber to gather a sample, of which nearly 60-90 % of all $N_2O$ fluxes were above the minimal detectable

flux (Fig. 5), we are confident that the uptake during dark measurements shows an actual biochemical process in the sampled sub-Arctic ecosystem, rather than an artefact. However, we acknowledge the potential for unknown chamber artefacts that may remain undiscovered and could affect the interpretation of our data.

**Table 1.** Comparison of chamber closure times for both light and dark measurements, with SE = Standard Error, and % = percentage difference between the absolute mean flux of chamber closure time x compared to chamber closure time 10 min.

| Chamber closure time | Light / Dark | Mean flux | SD flux | n | SE | % |
|---:|---|---:|---:|---:|---:|---:|
| 3 | light | -0.00 | 1.54 | 169 | 0.12 | 98.97 |
| 3 | dark | -1.49 | 1.80 | 170 | 0.14 | -15.16 |
| 4 | light | 0.20 | 1.26 | 169 | 0.10 | 43.58 |
| 4 | dark | -1.51 | 1.53 | 170 | 0.12 | -16.92 |
| 5 | light | 0.26 | 1.06 | 169 | 0.08 | 27.52 |
| 5 | dark | -1.47 | 1.39 | 170 | 0.11 | -14.18 |
| 6 | light | 0.30 | 0.94 | 169 | 0.07 | 16.21 |
| 6 | dark | -1.43 | 1.29 | 170 | 0.10 | -10.75 |
| 7 | light | 0.33 | 0.89 | 168 | 0.07 | 9.41 |
| 7 | dark | -1.39 | 1.24 | 170 | 0.10 | -8.05 |
| 8 | light | 0.35 | 0.83 | 168 | 0.06 | 2.35 |
| 8 | dark | -1.35 | 1.21 | 170 | 0.09 | -4.92 |
| 9 | light | 0.36 | 0.81 | 168 | 0.06 | 1.00 |
| 9 | dark | -1.34 | 1.16 | 170 | 0.09 | -3.79 |
| 10 | light | 0.36 | 0.80 | 168 | 0.06 | 0.00 |
| 10 | dark | -1.29 | 1.07 | 170 | 0.08 | 0.00 |

Mean flux rates change with chamber closure times because $N_2O$ builds up or decreases in the chamber head space. As the closure time increases, the concentration gradient between the soil and the head space decreases, reducing the diffusive flux.

Within a closed system, gases can reach a temporary state of equilibrium over time, where the rate of $N_2O$ production in the soil balances with the rate of $N_2O$ release into the atmosphere. Our results suggest that for light measurements, a chamber closure time of 3 min is too short and may result in significantly different flux rates than longer chamber closure times (ANOVA, p < 0.05 for 3 min compared to 8, 9, and 10 min). This discrepancy may be attributed to low microbial activity or the possibility that





$N_2O$ production is countered by its rapid uptake or dissolution in the water present in the soil matrix, a phenomenon previously

observed for $CO_2$ (Widén and Lindroth, 2003). From 4 min onward, mean $N_2O$ flux rates are not significantly different from one another (data not shown). However, it seems that with a chamber closure time longer than 7 min, the estimated $N_2O$ flux is less sensitive to increase during the chamber closure time, with fluxes 10%, 2% and 1% higher between 7 and 10 min compared to 10 min, respectively (Table 1). The proportion of light fluxes exceeding the minimum detectable flux (MDF) increased from 62.7% at a 3-minute closure time to 78.6% at a 10-minute closure time (Fig. 5 a) ). This trend suggests that while longer closure

times reduce the proportion of fluxes below the MDF and decrease variability, the gain between 4 and 10 min is marginal, with only 5% of fluxes failing to justify significantly prolonged chamber closure duration. It is likely that more observation points and higher concentration change renders the slope of the regression less sensitive to the noise. We thus suggest that for light measurements, depending on chamber size, chamber closure times of more than 4 min lead to reliable $N_2O$ flux estimates, with shorter chamber closures having the benefit of disturbing the soil profile for a shorter time.

For dark measurements, we find that the $N_2O$ uptake is highest with short chamber closure times, with flux rates around 15% lower at 3 - 5 min than at 10 min, respectively (Table 1). At 6 min, $N_2O$ uptake was still 10% higher than at 10 min, decreasing to below 8% between 7 and 9 min. At the same time, the MDF increased from 56.5% to 87.1 between 3 min and 10 min (Fig. 5 b) ). Nevertheless, none of the flux rates across different closure times were significantly different from one another (Kruskal-Wallis, p = 0.99). For dark measurements, we thus suggest to keep the chamber closure time between 3-5 min, unless

only few data points are available, when aiming for fluxes above the MDF becomes more important.

  With the goFlux output, we obtain an individual MDF for each measurement period, allowing us to determine on a case-by-case basis whether a flux is above the MDF. For both light and dark measurements, the MDF in our study was, on average, $0.015 \pm 0.001\,\mu\text{mol m}^{-2}\text{h}^{-1}$. This is lower than the reported $0.18\,\mu\text{mol m}^{-2}\text{h}^{-1}$ MDF rates in other nutrient-poor ecosystems by Christiansen et al. (2015) (Table 1).

While chamber measurements are essential for understanding GHG emissions, they can alter soil-air conditions and introduce observational artifacts. These include potential impacts such as pushing atmospheric air into the soil when closing the chamber, flushing soil gas into the chamber head space, and changing conditions during closure, *e.g.*, increase air temperature and humidity due to soil and plant evaporation (Subke et al., 2021; Rochette and Eriksen-Hamel, 2008). As a result, the $N_2O$ concentration gradient between the soil and the chamber head space and potentially also the functioning of plants and soil

microbes are altered and may cause a bias in the flux estimates (Davidson et al., 2002). At our site, condensation within the chamber during a measurement period increased drastically with time, especially in the colder months. Although our laboratory tests showed that for the Aeris-$N_2O$, increased water vapour does not interfere with $N_2O$ concentrations, all laser cells are sensitive to water vapour. Too high water vapour contents can, even with a filter assembly (1 micrometers pore size) within the tube, reach the analyser cell and lead to an abrupt end of a field campaign (Fiedler et al., 2022). Because this is well known,

goFlux takes the water vapour changes during a measurement period into account (Rheault et al., 2024). Nevertheless, it is crucial to know how water vapour increase over time during chamber closure. At our study site, $H_2O$ concentrations during light measurements increased, on average, from below 10000 ppm up to >16000 ppm, depending on the season. When we look at the rate of change over each minute, *i.e.*, 0-1 min, 1-2 min, 2-3 min etc., we can see that this rate of change drastically



**Figure 5.** Mean $N_2O$ *fluxes* (note: not concentrations) for light and dark measurements, with number of measurement periods above the minimal detectable flux (MDF, %). Note the different y-axes for the upper and lower plots.





decreases within the first 2 min, and then exponentially decreases with increasing chamber closure time (Fig. S5). In other
words, $H_2O$ concentrations rise drastically in the first 2 min (> 1300 ppm; data not shown), after which the level out until
around 8 min, before they start increasing again (Fig. S5). For dark measurements, the impact follows the same pattern, but is
of much smaller magnitude (approx. 600 ppm rise within the first 2 min; data not shown). However, it is also evident that the
increase of $H_2O$ concentrations does not seem to directly affect $N_2O$ concentrations in our study (Fig. S5).

   When transparent chambers are used, there is an additional constraint: the temperature within the chamber. Ideally, this
temperature stays constant during the whole measurement period; however, without any cooling systems attached, this is im-
possible to achieve in the field during sunny conditions as the chamber acts like a greenhouse (Fiedler et al., 2022). It is
suggested that these temperature increases can lead to an increased microbial activity leading to $N_2O$ production or consump-
tion, resulting in biased $N_2O$ concentrations (Rochette and Eriksen-Hamel, 2008; Clough et al., 2020). In our study, we can
see a similar pattern for the rate of change of chamber temperatures over time than with the $H_2O$ concentrations: the strongest
decrease happens within the first minutes of the measurement period (Fig. S5 and S6). At our site, temperature increased by
around 0.7°C within the first two minutes, which already slows down to 0.3°C after 5 min (data not shown). During dark mea-
surements, temperature within the chamber even tends to decrease slightly, by max. 0.2°C in the first two minutes and below
0.1°C after three minutes (data not shown). It is likely that during light measurements, the abrupt temperature increase in the
first two minutes may impact $N_2O$ concentrations (Fig. S6); however, cooling systems also have drawbacks, *e.g.*, causing ad-
ditional condensation within the chamber (Fiedler et al., 2022). As a result, we decided to avoid cooling systems and consider
temperature changes in the final flux calculation.

   To minimise disturbances to the soil gas-atmosphere gradient and obtain flux estimates close to pre-deployment levels,
several researchers have recommended using short chamber closure times of around 5 minutes (Fiedler et al., 2022; Pavelka
et al., 2018; Venterea and Baker, 2008). Especially for $N_2O$ uptake, it is crucial to keep the chamber closure time as short as
possible, as the $N_2O$ availability and its diffusion into the soil are often the limiting factor (Liu et al., 2022). This process is
driven by the concentration gradient between the atmosphere and the soil: when the chamber is closed, the $N_2O$ concentration
in the chamber head space is decreasing as $N_2O$ is taken up into the soil. As a result, the uptake rate is also decreasing,
since there is less $N_2O$ in the head space. Consequently, long chamber closure times may underestimate the uptake of $N_2O$
molecules. Our analyses of the chamber closure time confirm this: during dark measurements and over all measurement periods,
we found that the uptake rate at 3-5 minutes were greatest, and decreased with every minute of added chamber closure time
(Fig. 5). In contrast to this, as mentioned above, more than 40% of the fluxes were below the MDF at 3-minute closure time,
confirming that these very short closure times may result in higher uncertainties of flux estimates due to fewer sample points
(Christiansen et al., 2015). It is, therefore, crucial to consider the precision of the instruments used in the field to identify
the best-suited chamber closure time. With the Aeris-$N_2O$, we recommend chamber closure times between 4 and 8 min for
$N_2O$ measurements in low nutrient ecosystems, depending on chamber size and micro habitat, but also the length of the field
campaign: the shorter the chamber closure time, the more repetitions are possible, resulting in a higher amount of observations
per chamber base position. For dark measurements, we suggest shorter chamber closure times of 3-5 min. These findings are





in line with other studies (Cowan et al., 2014; Kroon et al., 2008; Christiansen et al., 2015) but confirm, for the first time, that these recommendations are applicable to nutrient-poor ecosystems.

### 3.3 Impact of linear and non-linear flux calculation approaches on $N_2O$ fluxes

To facilitate understanding of how $N_2O$ concentration build-up or reduction in the chamber head space can result in different flux estimates, goFlux automatically produces scatter plots with defined criteria (Fig. 6). These outputs allow for visual control of all measurement periods; additionally, outputs with the pre-defined quality checks are automatically produced. Using "best.flux" from the goFlux package, 59% (n = 2560) of all $N_2O$ fluxes during the different chamber closure times were calculated using the HM model, *i.e.*, concentrations showed a non-linear change over time, whilst 41% (n = 1728) were a result of linear (LM) model, *i.e.*, concentrations showed a linear change over time. However, all of the 41% fluxes calculated with the LM model had no HM flux, meaning that the software did not calculate a non-linear flux because the HM model gave the same results as the LM model, and therefore favoured the LM model. In other words, all fluxes were either calculated using the HM model or resulted in the same values as the LM model. This has two possible reasons: the first is that the linearity might be an outcome of short measurement time and low flux, so that the non-linear model was reduced to a linear model. Linear and non-linear fittings to the concentration data, as a function of time during a chamber closure, are often considered to exclude each other. However, the non-linear fitting includes also the linear fitting as a special case. If we use a generic exponential function $ae^{-bt}$, where a and b are positive constants to be fitted, it is asymptotically reduced to the linear form when $b$ is small. Namely, the first three terms of the serial expansion are $a(1 - bt + (bt)^2/2)$, and when $b << 1$ it is reduced to $a(1 - bt)$, that is the linear form. The slope of the linear term is readily identified as $-ab$. If we calculate the slope of the original exponential function by taking the time derivative, it gives $-abe^{-bt}$. Expanding it as a series and taking only the first order term as $b << 1$, we again obtain $-ab$ as slope. As conclusion, the general (exponential) fitting is automatically reduced to the linear fitting and to its slope if the data points are distributed in a linear manner. Although not novel, this conclusion does not appear to be commonly recognised within the chamber community; with our results, we show that for $N_2O$ fluxes, using a non-linear flux calculation approach yielded better or, for low concentration gradients, identical results as the linear model. Nevertheless, there is a second reason that linear models are favoured in goFlux: the non-linear model's curvature was too large, leading to flux estimates over four times higher (with a g.factor = 4) than those from the linear model. When we used a lower g.factor of 1.25 for comparison, *i.e.*, allowing HM fluxes to be max. 25% higher than LM, we found that about one-fourth of the fluxes estimated by the non-linear model would have been excluded due to significant overestimation.

There is a tendency to favour LM over HM models in literature, primarily due to its simplicity. It is also generally assumed that concentration changes are linear during short chamber closure times, keeping uncertainties low (Hüppi et al., 2018; Kroon et al., 2008). However, because GHG concentrations naturally follow a non-linear behaviour within a closed system due to diffusion theory (Fick's first law) and leakage (Anthony et al., 1995; Kroon et al., 2008), LM models may introduce a bias, differing from HM model estimates by up to 60% (Hüppi et al., 2018; Kroon et al., 2008), resulting in less accurate flux estimates and GHG budgets. This has been thoroughly investigated for $N_2O$ fluxes by Kroon et al. (2008), who found a large underestimation of $N_2O$ fluxes by the LM model in their study. With our results, including the mathematical explanation


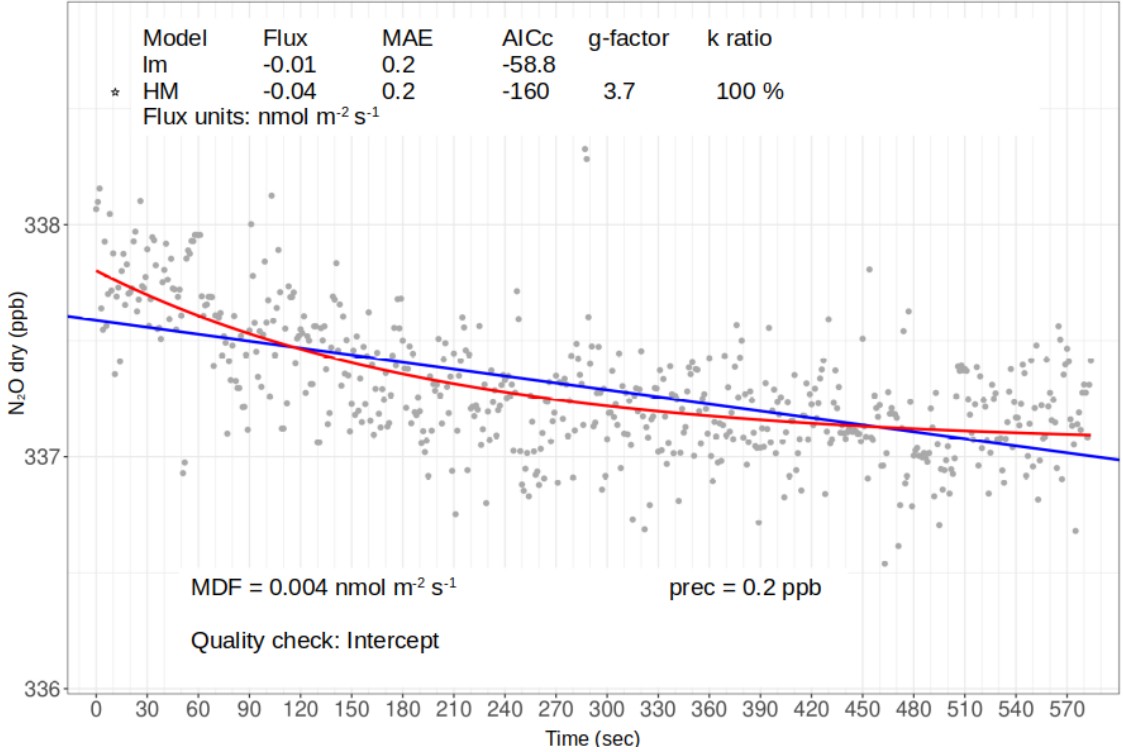

**Figure 6.** Example of scatterplot output from goFlux, showing $N_2O$ concentrations in ppb from one measurement period, with information on linear (LM, blue line) and non-linear (HM, red line) flux calculations. For every measurement period, the chosen model is marked with a star (here, HM) according to the pre-defined quality check, indicated on the bottom of the graph (here, intercept). Flux values, mean absolute error (MAE), g-factor and k ratio are shown on top of the figure; here, a g-factor of 3.7 indicates that the HM flux value is 3.7 times higher than the one obtained from LM. Source: goFlux package (Rheault et al., 2024), font sizes modified.

stating that LM models are obtained from HM models, we suggest that all future $N_2O$ flux chamber calculations should be done using HM models, which can be filtered for overestimation of fluxes when flux rates are larger. Novel software packages such as goFlux (Rheault et al., 2024) offer the possibility easily integrate both LM and HM models, and report the flux rates in
a reproducible way. We believe that these approaches will be crucial to facilitate the use of both LM and HM models, and, as a consequence, enable the chamber community to standardise their flux calculation methods.

### 3.4 Simulated differences in $N_2O$ flux rates between GC and PGA

Our results show that for light measurements, $N_2O$ fluxes obtained with three simulated GC- samples were, on average, 21.7% (0.085 $\mu$g $N_2O$-N m$^{-2}$, h$^{-1}$; data not shown) lower than PGA fluxes; especially positive fluxes were generally underestimated
(p $<$ 0.001, $R^2_{adj} = 0.37$, Fig. 7). Using four simulated GC- samples, negative $N_2O$ fluxes appeared to be nearly identical with the $N_2O$ fluxes gathered by PGA; however, positive fluxes were still underestimated by 3% (p $<$ 0.001, $R^2_{adj} = 0.65$). Interestingly,





by increasing the sample size to five or six, negative GC-simulated $N_2O$ fluxes seemed to be underestimated, whilst positive ones were overestimated ($p < 0.001$, $R^2_{adj} = 0.85$, Fig. 7). Overall, GC-simulations with five samples underestimated $N_2O$ fluxes by 2.2%, while simulations with six samples resulted in an overestimation of around 2.4% (data not shown).

**Figure 7.** Simulated GC fluxes from light measurements with a) 3, b) 4, c) 5, and d) 6 samples compared to flux measurements taken with a PGA for 10 min (n = 168). All fluxes were calculated with the goFlux package; results from "best.Flux" are shown. The blue trend line fits a generalised linear model, with the shaded area representing the 95% confidence interval. The shown equations and $R^2_{adj}$ values follow the $y \sim x$ equation.





For dark measurements, all $N_2O$ flux estimates gained from GC simulations were lower than the fluxes measured by the PGA, with an underestimation of 6.6%, 2.3%, 7.9%, and 8.1% for three, four, five, and six sample points (data not shown). With three samples, negative fluxes were generally overestimated, while positive fluxes were underestimated. With four sample points, negative fluxes still got overestimated, while with five and six sample points, negative fluxes were slightly underestimated. However, compared to light measurements, the $R^2_{adj}$ values were low (Fig. 8, $R^2_{adj}$ = 0.21, 0.51, 0.69, and 0.68,
respectively).

Our results are somewhat similar to previous studies comparing $N_2O$ flux rates between GCs and PGAs. These studies found that GCs generally underestimated $N_2O$ fluxes compared to fast-responding analysers, but concluded that GCs were still suitable for measuring $N_2O$ fluxes (Christiansen et al., 2015; Brümmer et al., 2017). Christiansen et al. (2015) investigated the differences between a fast-responding analyser (a cavity ring-down spectroscopy analyser (Fleck et al., 2013)) and a GC in
agricultural fields in Vancouver (Canada), with five GC samples at 0, 3, 10, 20, and 30 min chamber closure time. They found that $N_2O$ fluxes were very similar and did not differ significantly, with average $N_2O$ fluxes of 47.6 ± 8.4 $\mu$g $N_2O$-N m$^{-2}$ h$^{-1}$ from the the fast-responding analyser and 61.6 ± 11.2 $\mu$g $N_2O$-N m$^{-2}$ h$^{-1}$ from GC, respectively. With a similar setup, Brümmer et al. (2017) compared $N_2O$ fluxes measured by a fast-responding analyser similar to the Aeris-$N_2O$ (a quantum cascade laser) and a GC from an low-flux agricultural grassland in Braunschweig (Germany). Their four GC samples taken
at 0, 20, 40, and 60 min were highly scattered and rarely showed a distinctive trend, introducing a wide range of $N_2O$ fluxes between -26 to 39 $\mu$g $N_2O$-N m$^{-2}$ h$^{-1}$ with a standard error between 1 and 44 $\mu$g $N_2O$-N m$^{-2}$ h$^{-1}$. In contrast, the $N_2O$ fluxes measured by the fast-responding analyser only varied between 4 and 32 $\mu$g $N_2O$-N m$^{-2}$ h$^{-1}$, with standard errors below 1.2 $\mu$g $N_2O$-N m$^{-2}$ h$^{-1}$. It is likely that our results differ because of the short chamber closure time available for our GC simulation, with 10 min compared to 30 min and 60 min in the other studies. During prolonged chamber closure times,
substantial changes in the concentration gradient and chamber conditions can take place, which are unlikely to happen in the same extent in our GC-simulation. This highlights three critical aspects: first, despite claiming low-flux environments, flux rates from agricultural fields are much higher than from a sub-Arctic peatland or other nutrient-poor ecosystems (Savage et al., 2014; Cowan et al., 2014), where capturing $N_2O$ fluxes is even more challenging. Second, low $N_2O$ fluxes tend to be very scattered, which large noise in comparison to the actual trend, *i.e.*, the change in concentration during chamber closure. This
makes it very challenging to find trends when calculating fluxes if only few samples are available, let alone showing $N_2O$ uptake without high uncertainties (Cowan et al., 2014). Finally, it is crucial to know and test the precision of the instruments used in the field to get reliable flux estimates and minimal detectable fluxes.

We suggest that measuring $N_2O$ fluxes with fast-responding analysers in nutrient-poor ecosystems should be considered for all future studies. PGAs, for example, have two main advantages over the GC method: they collect a large amount of sample
points, and the quality of these can be checked *in situ* during the measurement period. With more samples, there are more data points, which then results in the option to reduce chamber closure times and further reduce artefacts caused by sealing off a part of a soil profile by a closed chamber (Brümmer et al., 2017). If errors happen in the field, *e.g.*, atmospheric air leaking into the chamber due to not properly closing it, or abrupt pressure changes when closing the chamber to brusquely, they are visible in the online interface of the PGA. This real-time *in situ* control of $N_2O$ concentrations allows for direct optimisation in the



Dark GC simulations with 3 samples (n = 169)

**a)**
$$y = -0.058 + 0.91\,x \quad R^2_{\text{adj}} = 0.21$$
$$p < 0.001$$

Dark GC simulations with 4 samples (n = 169)

**b)**
$$y = 0.013 + x \quad R^2_{\text{adj}} = 0.51$$
$$p < 0.001$$

Dark GC simulations with 5 samples (n = 169)

**c)**
$$y = 0.0069 + 1.1\,x \quad R^2_{\text{adj}} = 0.69$$
$$p < 0.001$$

Dark GC simulations with 6 samples (n = 169)

**d)**
$$y = -0.015 + 1.1\,x \quad R^2_{\text{adj}} = 0.68$$
$$p < 0.001$$

GC N$_2$O flux ($\mu$g N$_2$O$-$N m$^{-2}$ h$^{-1}$)

PGA N$_2$O flux ($\mu$g N$_2$O$-$N m$^{-2}$ h$^{-1}$)

**Figure 8.** Simulated GC fluxes from dark measurements with a) 3, b) 4, c) 5, and d) 6 samples compared to flux measurements taken with a PGA for 10 min (n = 338). All fluxes were calculated with the goFlux package; results from "best.Flux" are shown. The blue trend line fits a generalised linear model, with the shaded area representing the 95% confidence interval. The shown equations and $R^2_{\text{adj}}$ values follow the $y \sim x$ equation.





field and increases the quality of flux measurements (Fiedler et al., 2022). A practical result of that is that measurement periods can be interrupted and repeated in the field at any time, ensuring high quality of the flux measurements, as well as an optimal use of minimal time in the field, particularly since chamber closure times with PGAs are shorter than with GCs. In addition, most studies using GC have, so far, focussed on soil respiration by only conducting dark measurements; consequently, there is a lack of data on how soil $N_2O$ fluxes differ in light and dark conditions, especially in Arctic ecosystems (Stewart et al.,

2012), which should be investigated in future studies. However, PGAs have some drawbacks: weighing 10-20 kg (including batteries), they are heavier than GC vials. Additionally, their power consumption requires regular backups, and factors such as heavy vibrations, particles, water, and sudden pressure changes can contaminate the laser cell (Fiedler et al., 2022). With good planning and care, it is, however, easily possible to deal with these disadvantages.

It is important to note that our comparison was made between the PGA and simulated GC measurements. For the GC

simulations, we adjusted the instrument precision during the flux calculation, but no actual air samples were analysed by any GC instrument. Furthermore, as mentioned above, our chamber closure time was considerably shorter than for most GC studies because of the condensation and temperature changes within the chamber. Thus, keeping the chamber closure time to max. 10 min proved to be the best solution in our study. Nevertheless, our sensitivity analysis with 4 simulated GC samples showed that even when the sample times changed to $\pm$ 30 sec compared to the original time stamp, flux rates differed less than 10%, with

$R^2_{\text{adj}}$ values between 92 and 98 (data not shown). We believe that the underestimation of our $N_2O$ flux rates is, therefore, not a result of a inadequate simulation, but needs to be verified by future studies actually measuring $N_2O$ samples from nutrient-poor ecosystems in a GC. We strongly advise against using only 3 samples, as flux calculations may have to be discarded if only one sample is erroneous. Undetected errors can also bias flux estimates due to the high impact of each data point on the regression slope. To compare previous $N_2O$ flux measurements to novel data sets measured with a PGA, it is crucial to

investigate differences between these methods. To achieve this, novel instruments have to be tested on their precision, noise, and accuracy, as well as potential interference with water vapour (Grace et al., 2020; Ahmed et al., 2024).

## 4  Conclusions

The primary aim of this study was to establish a flux chamber method capable of quantifying low $N_2O$ fluxes in nutrient-poor ecosystems. Using a portable $N_2O$ analyser (Aeris Technologies; sensitivity: 0.2 ppb/s for $CO_2$ and $N_2O$, 1 Hz frequency) and

both transparent and dark flux chambers, we generated a first, extensive dataset of $N_2O$ fluxes in the (sub-) Arctic. Our laboratory tests confirmed the Aeris-$N_2O$'s suitability for measuring low $N_2O$ fluxes, with low noise, minimal water interference, and negligible signal drift. Our comparison of chamber closure times (3–10 min) showed that 3 minutes may be insufficient for capturing low $N_2O$ fluxes during light measurements, while closure times of 4–10 minutes provide more reliable results, with shorter chamber closures having the benefit of disturbing the soil profile less. For dark measurements, $N_2O$ uptake was

highest with short closure times. We recommend a chamber closure time of 3–5 minutes unless field measurement time data are limited, in which case longer times may help capture fluxes above the MDF. These recommendations should be adjusted based on chamber size, instrument precision, study site, and field campaign duration: shorter closure times allow more repetitions,





increasing the number of observations per chamber position. In our study, all $N_2O$ fluxes were calculated using the non-linear (HM) model or matched the linear (LM) model when data showed a linear distribution. This occurs because non-linear fitting

defaults to linear fitting when concentrations and slopes are low. While not new, this concept is often overlooked in the chamber community. Our results demonstrate that the HM model provides the best results or aligns with the LM model for low concentration gradients. We recommend using HM models for future flux calculations, with filters to address overestimation at higher flux rates. Novel software packages such as goFlux (Rheault et al., 2024) simplify the integration of both LM and HM models, and report flux rates in a reproducible way; such approaches are key to standardising flux calculations across the chamber

community. Our simulated comparison between the Aeris-$N_2O$ and GC showed that the GC generally underestimated $N_2O$ fluxes. Due to the high variability of $N_2O$ fluxes, we recommend using fast-responding analysers in nutrient-poor ecosystems. Fast-responding analysers collect more data points and allow real-time *in situ* quality control, enabling optimisation and quality of flux measurements in the field (Fiedler et al., 2022).

Our study site, a thawing permafrost peatland in sub-Arctic Sweden, acted as a sink of $N_2O$, with a mean flux of -0.469 $\pm$

1.25 $\mu$g $N_2O$-N m$^{-2}$ h$^{-1}$ during a chamber closure time of 10 min. Light and dark measurements gave 0.361 $\pm$ 0.797 and -1.29 $\pm$ 1.07 $\mu$g $N_2O$-N m$^{-2}$ h$^{-1}$ during a chamber closure time of 10 min, respectively. With 388 measurement periods for both light and dark measurements, of which up to 90 % of all $N_2O$ fluxes were above the minimal detectable flux, we are confident that this uptake reflects a real biochemical process rather than an artefact. While this study concentrates on the methodological aspects of quantifying $N_2O$ fluxes in a nutrient-poor ecosystem, a follow-up study will investigate the environmental drivers

of $N_2O$ fluxes. Our results demonstrate that low $N_2O$ fluxes can be measured in the Arctic, with notable differences between light and dark conditions that require further investigation. This fills an important gap in $N_2O$ studies from the Arctic, where negative fluxes have been observed, but could not be investigated due to measurement accuracy not being high enough (Voigt et al., 2020). It also highlights the need for future research on $N_2O$ fluxes in sub-Arctic ecosystems and other nutrient-poor ecosystems, and their potential response to global warming.

*Code availability.* The scripts for processing and analysing the data are publicly available at https://git.bgc-jena.mpg.de/ntriches/data-analysis/-/tags/2024-12-12-triches-amtsubmission-n2oadvances under the terms of the GNU General Public License version 3.

**Appendix A**

Sampling ambient air using Aeris-$N_2O$ analyser, Fig. A1 shows the entire 15 hours long run without removing the first 5 hours (1 Hz), where the water vapour mole fractions were not stable.



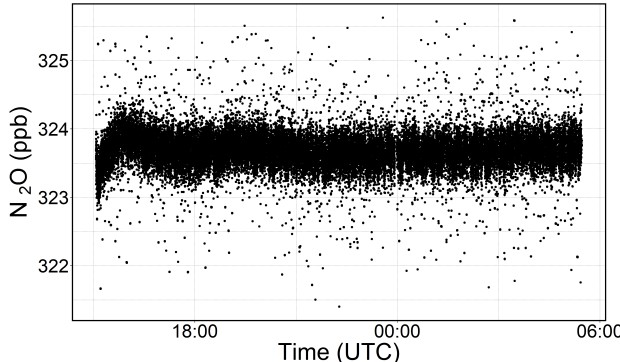

**Figure A1.** Time series of the $N_2O$-Analyser of 15 hrs long run.

Changes of $H_2O$ mole fractions during the water interference test (see Fig. A2). Corresponding relative humidity values were noted and the data used for the comparison were designated with red vertical lines.

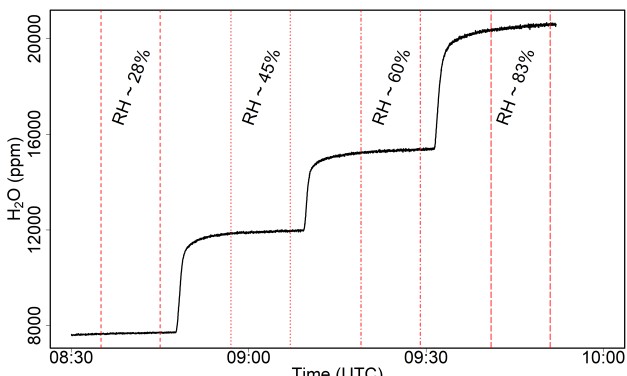

**Figure A2.** $H_2O$ mole fractions measured by Aeris-$N_2O$ analyser during the water interference test. The corresponding relative humidity (RH) values were given for each step. Red lines demonstrate the data used to assess the water interference at each RH steps.



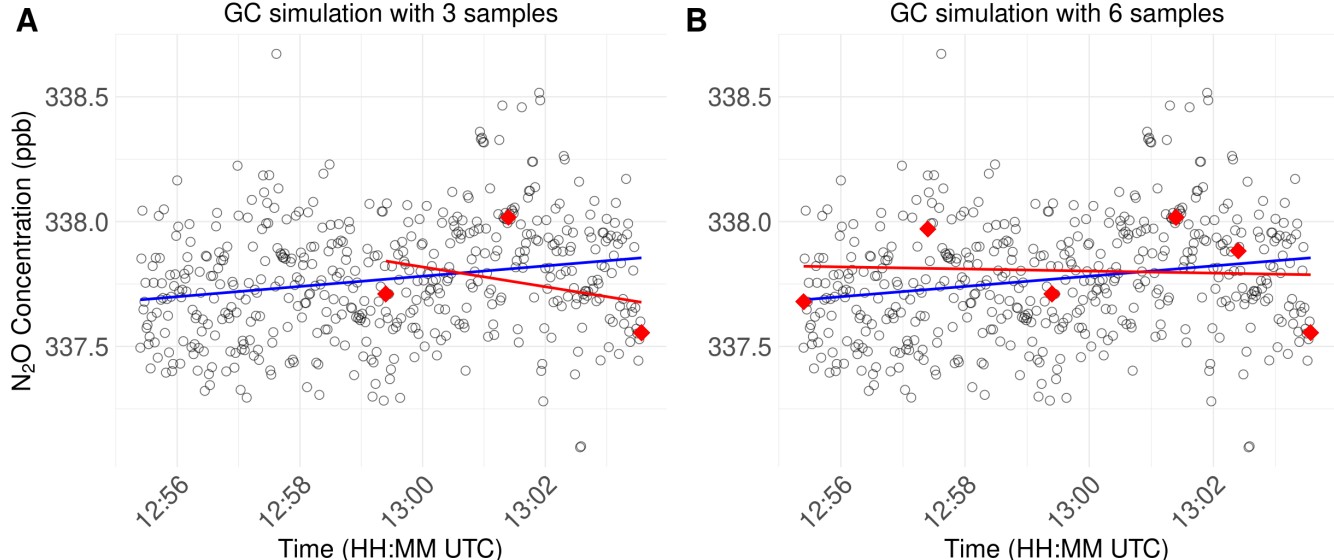

**Figure A3.** Examples visualising the comparison of regression slopes obtained using 600 data points from the portable gas analyser (blue line) vs 3 (A) or 6 (B) virtual samples mimicking manually defined sample times for subsequent analysis in a gas chromatograph (red line), respectively.

*Author contributions.* NYT designed the experiment, collected, and processed the data, and did the laboratory test together with AB. AB further analysed and reported the laboratory data, and created the QGIS figures. NYT and JE developed and implemented the scripts used for data processing, quality checks, analysis, and GC simulations. TV wrote the mathematical explanation of non-linear and linear models. NYT wrote the first draft of the manuscript, and AB, AMV, MEM, MG, MH and TV provided valuable comments that helped improving it. NYT was supervised by MG, MH, TV, AMV, and MEM. MH and MG were responsible for funding acquisition.

*Competing interests.* The authors declare that they have no conflict of interest.

*Acknowledgements.* The presented research was supported by the European Research Council (ERC) under the European Union's Horizon 2020 research and innovation programme (grant agreement No 951288, Q-Arctic) and by ICOS-Finland (University of Helsinki).The work of MEM was financed by Research Council of Finland-funded projects Thaw-N (no. 353858) and N-Perm (no. 341348). AMV acknowledges funding catalyzed by the TED Audacious Project (Permafrost Pathways).

The authors thank the 'Field experiments & instrumentation' and 'GasLab' service groups at the Max Planck Institute for Biogeochemistry for their help in designing the chamber system and testing the Aeris-$N_2O$. Further thanks to the field assistants Alena Markelova, Antonin Affolder, Mark Schlutow, Mirkka Rovamo, Valentin Kriegel and Wasi Hashmi, as well as the staff from the Abisko Scientific Research Station and Mattias Dalkvist. Many thanks to Karelle Rheault for her continuous help with the goFlux package, and Jesper Christiansen



for support in the interpretation of non-linear and linear flux rates. Authors also thank Danilo Custódio at MPI-BGC/BSI for his valuable comments and suggestions which helped us to improve this manuscript.



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
