# Peer review of "Practical guidelines for reproducible N2O flux chamber measurements in nutrient-poor ecosystems"

_Atmospheric Measurement Techniques, 2024_

## Referee Comment (RC1)

**Review of "Advancing $N_2O$ flux chamber measurement techniques in nutrient-poor ecosystems"**

Triches et al. 2025, Atmospheric Measurement Techniques

**Overview**

This study investigates the quantification of low nitrous oxide ($N_2O$) fluxes in nutrient-poor, sub-Arctic ecosystems using a fast-responding portable gas analyser (PGA, Aeris-$N_2O$) combined with a custom-built flux chamber system. The authors conducted laboratory tests to assess the instrument's signal stability, noise characteristics, and water vapor interference, and performed field experiments at a sub-Arctic peatland. Chamber closure times ranging from 3 to 10 minutes were systematically varied under both light and dark conditions, and fluxes were calculated using both linear and non-linear models. In addition, the high-frequency PGA data were downsampled to simulate gas chromatograph (GC) measurements, and the impact of reduced sampling frequency on parameter uncertainty and flux estimates was evaluated.

I would like to thank the authors for providing the data and code for this study.

The paper is well written and is suitable for publication in Atmospheric Measurement Techniques. I have three general comments regarding the uncertainty analysis, the simulation of GC measurements, and the model assumptions.

**General comments**

1. The analysis of the instrument stability and noise characteristics seems to suffer from repetition of several values that are very close to each other (see figure below). As a result, the Allan deviation analysis shows correlated noise up to 30 seconds averaging period. When the repeated values are removed, the Allan deviation shows a more realistic white noise behavior. These repeated values need to be removed before any further analysis. Note that they are not identical but they are very close to each other.

2. The comparison with GC simulation is not rigorous enough in my opinion. Report-

[Figure]

Figure 1: A random sample of 2 minutes from the lab test time series

[Figure]

Figure 2: Corrected Allan deviation after removing duplicates

ing underestimation from a downsampled time series needs stronger justification. An increase in uncertainty from a fewer number of sampling points is expected and can be assessed from theory. That being said, a Monte Carlo simulation is also a valid approach to explore the increased uncertainty. However, in that case the subsampled time series should be taken from what is believed to be the true value of concentration (i.e. from the fitted model) and then noise can be added to it based on known GC uncertainty e.g. (Lebegue et al. 2016) (Rapson and Dacres 2014). Then the fluxes calculated from simulated GC measurements can be compared to PGA measurements.

If the authors' argument is that there is more than just downsampling and that the concentration inside the camber fluctuates not only due to noise, then a justification needs to be provided for this argument. In that case taking mean downsampled PGA measurements is reasonable but needs explicit discussion of the uncertainty.

3. It appears that both the LM and HM models assume that flux at the soil-air interface remains constant over time (time invariant). However, the study mentions that $N_2O$ fluxes change as the concentration gradient evolves within the chamber. Could the authors discuss this assumption more explicitly and its potential implications for flux calculations? Are there metrics that the authors recommend to assess the violations of this assumption?

**Specific comments**

(*numbers refer to line numbers in the manuscript*)

- The title of the paper seems quite general. You might consider suggesting a more specific title that highlights the key contributions but I leave this to the authors to decide.

- 12 ... "we **can** successfully detect" This sounds like you are reporting a detection limit. but you are reporting means with an uncertainty interval. I would suggest to rephrase this sentence.

- 15 MDF reported as a number with uncertainty interval is unusual. Maybe clarify what is reported here.

- 20 "..or matched the linear model. .." could you clarify this sentence.

- 21 Underestimation flux when using GC needs stronger justification see my general comment.

- 27 is the repetition of the IPCC full name necessary?

- 57 "At the same time, many of the reported $N_2O$ fluxes were found to be below the detection limit, making it challenging to assess the magnitudes and possible uptake of these fluxes."

Not clear here which fluxes you are referring to. The phrase "possible uptake" sounds ambiguous to me. do you mean "challenging to assess the sign of these fluxes"?

- 91 the coordinates seem to be missing a dot "20.0'" instead of "200"

- 180 where is the instrument precision taken from?

- 180 Definition of MDF is somehow strange as it does not include the number of points in the time series. Also this definition is different from what what is typically reported in the literature. For example (Maier et al. 2022) defines MDF as

$$MDF = \left(\frac{PI}{t_c}\sqrt{\frac{t_c}{p_s}}\right) \times \left(\frac{V_{ch}p_{air}}{ART_{air}}\right)$$

- I think it's also important to highlight that MDF defined here is only a theoretical limit based on the instrument uncertainty and does not account for model fit or chamber artifacts.

- 194 "We used this factor because it has been previously used" I think better justification is needed here.

- 202 could you clarifies your second measurement strategy? did you take block averages or have you calculated the flux on 1-min intervals?

- 223 2500 ppm to 800 ppm seem very dry for ambient air, was their any drying involved?

- Figure 1 description needs restructuring. Move information about shape files to the end if it is needed. Also shapefile is a technical term that might not be familiar to all readers. I would suggest to use a more general term like "geospatial data" or "map data" instead.

- Looking at Figure A1. I would not say there is no drift. At least in the beginning (maybe 2, 3 hours) there is a clear drift of about 1.5 ppb. You mentioned that 5 hours were needed for the signal to stabilize, would that mean the instrument needs 5 hours in the field for the signal to stabilize? or is that a temperature or pressure artifact? this needs more discussion.

- Figure 4. is there a reason why relative humidity is chosen rather than absolute water concentration? For the analyzer, any matrix effects would be dependent on absolute water concentration rather than relative humidity.

- 240 with such large uncertainty - this is basically not statistically different from zero - I would not conclude that the site acted as a sink. What is the number after the $\pm$ in the reported fluxes? Conventionally, it should be 2*SE to give a 95% confidence interval for the mean (can you please mention this when your first report it)

- 245: is there something that supports this?

- 269: I don't understand the recommendation for 3-5 minutes since the flux is continuously increasing. Could you elaborate more on this recommendation?

- 291: why does water increase in that pattern, and why would it decrease with time? Does temperature explain this behavior?

- Figure S5, is this a differenced time series? so is this ppm per minute or is it just normalized concentration by subtracting the first H2O measurement.

- 378 again I think an explanation is needed on why GC underestimates the flux

- in the conclusion, can you extend your recommendations to cover model choice, and how to avoid chamber artifacts. I am thinking of what metrics would you recommend to assess the goodness of fit of the model or when to detect the violations of the time-invariance assumption.

**References**

Lebegue, Benjamin, Martina Schmidt, Michel Ramonet, Benoit Wastine, Camille Yver Kwok, Olivier Laurent, Sauveur Belviso, et al. 2016. "Comparison of Nitrous Oxide (N$_2$O) Analyzers for High-Precision Measurements of Atmospheric Mole Fractions." *Atmospheric Measurement Techniques* 9 (3): 1221–38. https://doi.org/10.5194/amt-9-1221-2016.

Maier, Martin, Tobias K. D. Weber, Jan Fiedler, Roland Fuß, Stephan Glatzel, Vytas Huth, Sabine Jordan, et al. 2022. "Introduction of a Guideline for Measurements of Greenhouse Gas Fluxes from Soils Using Non-Steady-State Chambers." *Journal of Plant Nutrition and Soil Science* 185 (4): 447–61. https://doi.org/10.1002/jpln.202200199.

Rapson, Trevor D., and Helen Dacres. 2014. "Analytical Techniques for Measuring Nitrous Oxide." *TrAC Trends in Analytical Chemistry* 54 (February): 65–74. https://doi.org/10.1016/j.trac.2013.11.004.

---

## Author Comment (AC1)

**Response to reviewer 2**

**Note:** Reviewer titles (overview, specific comments) are shown in **bold**. Reviewer comments (RC) are enumerated, with corresponding *author comments (AC) in italic.*

**Overview**

It is true that a lack of measurement precision so far prevents a sound in-situ assessment on the role of nutrient-poor ecosystems in $N_2O$ cycling and their potential to consume $N_2O$. E.g. in Huth et al. (2022) Restoration Ecology, 30, e13490, we found a tendency of $N_2O$ uptake in Sphagnum-moss dominated plots, but due to the low precision of the GC sampling and measurements, we could not determine if that was actually significantly different from 0 or not. Therefore I much appreciate the efforts made by Triches et al. to test high-precision $N_2O$ measurements in a low-flux environment as this will substantially help elucidating the role of northern nutrient-poor ecosystems in global $N_2O$ cycling. The manuscript is generally well-written and fits nicely into the scope of the journal Atmospheric Measurement Techniques. I only had some minor comments and suggestions to make (especially in the discussion), except for the point that at very low $N_2O$ fluxes, $CO_2$ uptake or efflux due to transparent or non-transparent chamber measurements could actually become a factor in either enriching or diluting $N_2O$ during closure time. Since I'm wondering if this may explain the differing results between the transparent and non-transparent chamber measurements I would encourage the authors to use the Aeris $CO_2$ data and check, if a correction similar to water vapor would change the results. If $CO_2$ data is not available, I believe this should at least be thoroughly discussed.

*We thank Dr. Huth for the kind words and thorough review, as well as for providing some relevant references that we had missed. We agree that possible interference of $N_2O$ and $CO_2$ fluxes during the chamber closure time may cause issues. We know that tests on the Aeris Mira Ultra $N_2O$ $CO_2$ analyser have been conducted in 2023, showing a significant $N_2O$ $CO_2$ crosstalk with a decrease of -0.008 ppb $N_2O$ per ppm $CO_2$. Considering that the change in $CO_2$ concentrations we measured in the field are generally around 60-80 ppm with a simultaneous decrease of 0.6-0.7 ppb of $N_2O$ (20 x 0.008 = 0.16), this suggests that the difference between dark and light cannot only be explained due to the sensors' interference. We have also analysed our $N_2O$ and $CO_2$ data thoroughly and have observed some trends. However, we hope Dr. Huth will understand that*

*this is a discussion for another manuscript, which is indeed in preparation. For the follow-up manuscript, we will, as suggested*

*by Dr. Huth, add a small laboratory analysis to check how realistic $CO_2$ concentration changes affect the $N_2O$ concentrations*

*in our instrument.*

**Specific comments**

Abstract:

RC 1. L.15-19: Please shortly mention your chamber height, because closure times are dependent on that.

*AC 1. Thanks, we will make sure to mention it. Please see our abstract in the revised manuscript.*

Introduction:

RC 2. L.33: Why not give credit to the early studies, e.g. Martikainen et al. (1993) Nature, 366, 51–53 or Nykänen et al. (1995)

Journal of Biogeography, 22, 351–357.

*AC 2. We agree, thank you. We will add the first suggested study as reference.*

RC 3. L.34: If N availability is low, $N_2O$ uptake might be expected (Buchen et al. 2019, Soil Biology and Biochemistry, 130,

63-72) but up to now it was extremely hard to detect, e.g. via Helium incubation studies (ibid). The value of this study to me is that the role of (northern) nutrient-poor ecosystems in $N_2O$ cycling and potential uptake could now be elucidated.

*AC 3. We will adjust the text accordingly in the introduction and conclusion: "Until about 15 years ago, only few studies*

*investigated $N_2O$ fluxes in the (sub) Arctic, where soils often have a very low availability of reactive N (Virkkala et al.,*

*2024) and thus are not expected to emit amounts of $N_2O$ relevant for the global climate (Voigt et al., 2020; Christensen*

*et al., 1999; Grogan et al., 2004; Martikainen et al., 1993). In these low N ecosystems, $N_2O$ uptake could be expected,*

*but has, so far, not been confirmed in field studies (Buchen et al., 2019; Schlesinger, 2013)."*

RC 4. L.61: Please add: "under a fixed chamber height", because closure times are directly depending on it (see Fiedler et al.

2022).

*AC 4. Thanks for the reviewer comment, we will revise the manuscript as suggested.*

RC 5. L.75: Was the chamber really dark? In general, the terms "light" and "dark" measurements can easily be misleading (e.g.

our non-transparent chambers/our shading tarps are usually white to increase reflection and reduce chamber heating and

I guess you did not really measure light, did you?), I would suggest you just use "transparent" and "non-transparent" (or

"opaque") measurements/chambers etc. throughout the manuscript.

*AC 5. Very good point. We only used transparent chambers; for the "dark" measurements, we indeed just added a*

*reflecting tarp on top of the chamber, which also covered the PAR sensor. We agree that transparent and opaque is more*

*easily understandable and inclusive, and will change the manuscript accordingly.*

2 Methods

RC 6. L.166: Does the Aeris analyser do not give dry mole fractions of the target gas? If so, shortly mention here or where you introduce the analyser.

*AC 6. It does. We will add this to the introduction of the analyser: "To measure $N_2O$ concentrations, we used the Aeris*

*MIRA Ultra $N_2O$/$CO_2$ (from now onward: Aeris-$N_2O$) analyser (Aeris Technologies; sensitivity: 0.2 ppb/s for $CO_2$*

*and $N_2O$, frequency: 1 Hz). As most PGA, the Aeris-$N_2O$ provides dry mole fractions of the target gas. We performed*

*several laboratory tests to assess the signal stability (i.e., drifts and stabilisation time), uncertainties, noise level, and*

*water interference of the Aeris-$N_2O$. The analyser was left to sample ambient air for approximately 15 hrs to evaluate*

*the signal stability (see section 3.1)."*

3 Results and Discussion

RC 7. L. 226: Does this warming period occur every time once you turn on the analyser or was this first-use? Does this have implications for field application? I did not quite get if it is a problem, because at different RH, $N_2O$ concentration seems to be stable. Please just quickly inform the reader, if the analyser warm up may pose a problem or not.

*AC 7. This warming period occurs every time the analyser is turned on. The implications for the field applications are*

*that we never turned off the instrument during the whole field campaign, which was recommended by the company and*

*is an important information for the reader. We will add this information as follows at the end of chapter 3.1: "Overall,*

*our conducted laboratory tests indicated that the Aeris-$N_2O$ was a suitable instrument for measuring low $N_2O$ fluxes,*

*showing low noise and water interference, along with negligible signal drift after the laser warms up. Nevertheless, the*

*long warm-up period (approximately 5 h) of the analyser needs to be taken into account, as this can be a limiting factor*

*for certain applications. To mitigate this, the Aeris-$N_2O$ remained powered on throughout the whole field campaign."*

RC 8.  240: It's called atmospheric sign convention and this could be stated already in the methods under your flux calculation procedure.

*AC 8. Thanks. We will add this after the flux calculation equation: "To report our flux rates, we used the atmospheric*

*sign convention, i. e., negative signs for an uptake of $N_2O$ into the soil, and positive signs for emissions."*

RC 9.  243-244: Complicated sentence and you already said in the methods, what a measurement period is. Please shorten and rephrase.

*AC 9. We agree and will rephrase the whole paragraph to: "At our site, we commonly observed net $N_2O$ consumption,*

*suggesting an atmospheric sink, with a mean flux of -0.469 $\mu g$ $N_2O$-N $m^{-2} h^{-1}$, with a 95% confidence interval (CI)*

*of (-0.60,-0.3) during a chamber closure time of 10 min. Our calculated mean flux during transparent measurements*

*was 0.361 $\mu g$ $N_2O$-N $m^{-2} h^{-1}$, with a 95% CI of (0.24,0.48) during a chamber closure time of 10 min (Table ??). For*

*opaque measurements, our calculated flux was -1.29 $\mu g$ $N_2O$-N $m^{-2} h^{-1}$, with a 95% CI of (-1.45,-1.13), indicating*

*that our opaque measurements represent a real biochemical process, rather than an experimental artefact, in the (sub-*

*) Arctic ecosystem. Nevertheless, the impact of environmental drivers on $N_2O$ fluxes, including the transparent and*

*opaque measurements, is beyond the scope of this study. Overall, we collected 338 samples, with 60-90 % of $N_2O$ fluxes*

*above the detectable limit. We therefore also acknowledge the possibility of unknown chamber artefacts that may remain*

*undiscovered and could affect the interpretation of our data."*

RC 10. L.245ff: This is my major point that needs some attention and discussion: At low $N_2O$ concentration changes,$CO_2$

concentration changes due to respiration (non-transparent measurments: $N_2O$ gets diluted, indicates uptake) or net uptake (transparent measurements: $N_2O$ gets enriched, indicates efflux) might become a factor. Either recalculate fluxes with a

$CO_2$ correction (like for water vapour), or add a short paragraph on this topic. If it is easily doable for you, you might also check in the lab, how realistic $CO_2$ concentration changes affect $N_2O$ concentration and add it to the manuscript.

*AC 10. We thank Dr. Huth for the detailed description of his major point. For its discussion, we would like to refer to our*

*response to his overview comment (see above, lines 17ff).*

RC 11. L314ff: Yes, but it also could just indicate $CO_2$ saturation during closure time, hence a decrease in $N_2O$ dilution.

*AC 11. Please see our response to the overview comment (lines 17ff).*

RC 12. L317: It's not really due to fewer sampling points but rather due to the lower absolute change in concentration, that is below the one needed for flux detection see also Fiedler et al. (2022)

*AC 11. Correct. We will rephrase to: "In contrast to this, as mentioned above, more than 40% of the fluxes were below*

*the MDF at 3-minute closure time. This confirms that very short closure times can lead to higher uncertainties of flux*

*estimates because the concentration changes are too small to be accurately detected (Fiedler et al., 2022)."*

RC 13. L320: It is not the chamber size, it's (effective) chamber height or V:A-ratio, that is determining concentration change and measurement length.

*AC 13. Correct, thank you. We will rephrase to: "The optimal closure time depends on factors such as chamber height,*

*micro habitat, and the duration of the field campaign."*

RC 14. L321: Jungkunst et al. (2018) Journal of Plant Nutrition and Soil Science, 181, 7-11 actually assessed the trade off between reducing temporal accuracy of flux measurements to gain more spatial replicates. Might want to cite this here.

*AC 14. Thank you, we will add it.*

RC 15. L336: What do you mean by LM and non-LM models exclude each other? The common approach is that if the difference between the two is non-significant, the simpler model should be used.

*AC 15. Thank you for pointing out this formulation. We will adjust the whole paragraph as follows: "Linear and non-*

*linear models for concentration data during chamber closure may typically be seen as alternatives, not complementary*

*approaches. However, the non-linear fitting includes the linear fitting as a special case. When using a generic exponential*

*function $ae^{-bt}$ to fit data, where a and b are positive constants to be fitted, it can be approximated to a linear function*

*if the data points are distributed linearly. This is because the exponential function can be expanded as a series, and*

*when the rate constant b is small, the linear function dominates. Namely, the first three terms of the serial expansion are*

*$a(1 - bt + (bt)^2/2$, but when b is very small, i.e., $b << 1$, it is reduced to $a(1 - bt)$, which is the linear form. The slope*

*of the linear term is $-ab$; if we take the time derivative of the original exponential function to calculate the slope, it*

*gives $-abe^{-bt}$. When we expand it as a series and only take the first order term as $b << 1$, we again obtain the simplified*

*$-ab$ as slope. This means that if the data points are linear, the exponential fitting will automatically reduce to a linear*

*fitting with the same slope. With our results, we show that for $N_2O$ fluxes, indeed, flux estimates were reduced to the*

*linear model and yielded identical results as the non-linear model."*

RC 16.  L343: Again, I don't understand this statement. If the data does not show non-linearity, linear models are sufficient. In any case, this statement should be rephrased, because I don't get how you assess that the relation between exponential and linear models do not appear to be recognised within the chamber community.

*AC 16. We agree and refer to the AC 15 above.*

RC 17.  L350ff: That is true, but in theory, calculating fluxes from closed-chamber measurements actually assume non-disturbance conditions and non-linear models are a tool to calculate fluxes from disturbed measurements. Therefore it is also justifiable to reduce chamber closure time to the most linear part, because this signifies non-disturbance in your measurement.

In other words, if concentration change is significantly different from linearity, chamber closure time was too long (or it wasn't properly sealed etc.).

*AC 17. Thank you for this comment. We agree that these are the assumptions for closed-chamber measurements and we do*

*not question the practice of reducing chamber closure times to the most linear part; indeed, we encourage short chamber*

*closure times. However, several studies assessing these assumptions have concluded that chamber measurements are*

*sensitive to errors (e.g., (Kutzbach et al., 2007; Pavelka et al., 2018)). On top of that, there are only few data sets*

*addressing $N_2O$ fluxes from nutrient-poor ecosystems and the high spatial and temporal variability of these fluxes*

*makes it challenging to assess them (Butterbach-Bahl et al., 2013). In our data, there is no evidence for oversaturation in the head space, i.e., too long chamber closure times, but non-linearity could still be observed. We further argue that non-linearity can be a result of a concentration change of the flux only, which can be corrected with non-linear fitting. This is why we encourage future research to include non-linear flux calculation to their linear calculation, and, ideally, justify their model choice with some metric.*

RC 18. L365ff: Does accuracy increase due to the fact that more GC samples better represent non-linearity of the data? Please discuss!

*AC 18. Thanks for the comment. We will address this in this statement: "The underestimation of GC fluxes may occur as a result of a smoothed out curve: when only few data points are available, variations in curves are naturally reduced. Furthermore, the precision of our GC was 1.9 ppb compared to 0.2 ppb of the Aeris-$N_2O$, resulting in less accurate measurements of the $N_2O$ concentrations. This may lead to a loss of detail in the curve, particularly in the peak values of the $N_2O$ concentrations, which can result in underestimation of the flux."*

RC 19. L.394: which = with?

*AC 19. Thanks, we will revise the manuscript as follows: "Second, low $N_2O$ fluxes tend to be very scattered, with large noise in comparison to the actual trend, i.e., the change in concentration during chamber closure."*

Conclusion

RC 20. Much of this is a repetition from previous paragraphs. Consider shortening and focusing on the main outcomes and recommendations.

*AC 20. Thank you. We will revise the conclusion accordingly, so that it includes the main outcomes, such as*

– *the laboratory work, showing that with the Aeris-$N_2O$ and our manual chamber system, we are able to report very low $N_2O$fluxes,*

– *our recommended chamber closure times, which can differ for transparent and opaque measurements,*

– *reasons for using **all data points** when calculating flux estimates using the HM model.*

*We further aim to give recommendations on why PGAs should be considered for future $N_2O$ flux studies, and try to en-*

*courage future research on $N_2O$ fluxes in nutrient-poor ecosystems. Please see our conclusions in the revised manuscript.*

Appendix

RC 21. Figure A3: That's a really nice figure that I believe would be well-placed within the main text.

*AC 21. Thank you. We will place figure A3 within the main text.*

**References**

Buchen, C., Roobroeck, D., Augustin, J., Behrendt, U., Boeckx, P., and Ulrich, A.: High N2O Consumption Potential of Weakly Disturbed Fen Mires with Dissimilar Denitrifier Community Structure, Soil Biology and Biochemistry, 130, 63–72, https://doi.org/10.1016/j.soilbio.2018.12.001, 2019.

Butterbach-Bahl, K., Baggs, E. M., Dannenmann, M., Kiese, R., and Zechmeister-Boltenstern, S.: Nitrous oxide emissions from soils: how well do we understand the processes and their controls?, Philosophical Transactions of the Royal Society B: Biological Sciences, 368, 20130 122, https://doi.org/10.1098/rstb.2013.0122, number: 1621, 2013.

Christensen, T. R., Michelsen, A., and Jonasson, S.: Exchange of CH4 and N2O in a subarctic heath soil: effects of inorganic N and P and amino acid addition, Soil Biology and Biochemistry, 31, 637–641, 1999.

Fiedler, J., Fuß, R., Glatzel, S., Hagemann, U., Huth, V., Jordan, S., Jurasinski, G., Kutzbach, L., Maier, M., Schäfer, K., Weber, T., and Weymann, D.: BEST PRACTICE GUIDELINE: Measurement of carbon dioxide, methane and nitrous oxide fluxes between soil-vegetation-systems and the atmosphere using non-steady state chambers, 2022.

Grogan, P., Michelsen, A., Ambus, P., and Jonasson, S.: Freeze–thaw regime effects on carbon and nitrogen dynamics in sub-arctic heath tundra mesocosms, Soil Biology and Biochemistry, 36, 641–654, https://doi.org/10.1016/j.soilbio.2003.12.007, 2004.

Kutzbach, L., Schneider, J., Sachs, T., Giebels, M., Nykänen, H., Shurpali, N. J., Martikainen, P. J., Alm, J., and Wilmking, M.: CO$_2$ flux determination by closed-chamber methods can be seriously biased by inappropriate application of linear regression, Biogeosciences, 4, 1005–1025, https://doi.org/10.5194/bg-4-1005-2007, 2007.

Martikainen, P. J., Nykänen, H., Crill, P., and Silvola, J.: Effect of a Lowered Water Table on Nitrous Oxide Fluxes from Northern Peatlands, Nature, 366, 51–53, https://doi.org/10.1038/366051a0, 1993.

Pavelka, M., Acosta, M., Kiese, R., Altimir, N., Brümmer, C., Crill, P., Darenova, E., Fuß, R., Gielen, B., Graf, A., Klemedtsson, L., Lohila, A., Longdoz, B., Lindroth, A., Nilsson, M., Jiménez, S. M., Merbold, L., Montagnani, L., Peichl, M., Pihlatie, M., Pumpanen, J., Ortiz, P. S., Silvennoinen, H., Skiba, U., Vestin, P., Weslien, P., Janous, D., and Kutsch, W.: Standardisation of chamber technique for CO2, N2O and CH4 fluxes measurements from terrestrial ecosystems, International Agrophysics, 32, 569–587, https://doi.org/10.1515/intag-2017-0045, number: 4, 2018.

Schlesinger, W. H.: An Estimate of the Global Sink for Nitrous Oxide in Soils, Global Change Biology, 19, 2929–2931, https://doi.org/10.1111/gcb.12239, 2013.

Virkkala, A.-M., Niittynen, P., Kemppinen, J., Marushchak, M. E., Voigt, C., Hensgens, G., Kerttula, J., Happonen, K., Tyystjärvi, V., Biasi,

C., Hultman, J., Rinne, J., and Luoto, M.: High-resolution spatial patterns and drivers of terrestrial ecosystem carbon dioxide, methane, and nitrous oxide fluxes in the tundra, Biogeosciences, 21, 335–355, https://doi.org/10.5194/bg-21-335-2024, 2024.

Voigt, C., Marushchak, M. E., Abbott, B. W., Biasi, C., Elberling, B., Siciliano, S. D., Sonnentag, O., Stewart, K. J., Yang, Y., and Martikainen, P. J.: Nitrous oxide emissions from permafrost-affected soils, Nature Reviews Earth & Environment, 1, 420–434, https://doi.org/10.1038/s43017-020-0063-9, 2020.

---

## Author Comment (AC2)

**Response to reviewer 1**

**Note:** Reviewer titles (overview, general comments, and specific comments) are shown in **bold**. Reviewer comments (RC) are enumerated, with corresponding *author comments (AC) in italic*.

**Overview**

This study investigates the quantification of low nitrous oxide ($N_2O$) fluxes in nutrient- poor, sub-Arctic ecosystems using a fast-responding portable gas analyser (PGA, Aeris- $N_2O$) combined with a custom-built flux chamber system. The authors conducted laboratory tests to assess the instrument's signal stability, noise characteristics, and water vapor interference, and performed field experiments at a sub-Arctic peatland. Chamber closure times ranging from 3 to 10 minutes were systematically varied under both light and dark conditions, and fluxes were calculated using both linear and non- linear models. In addition, the high-frequency PGA data were downsampled to simulate gas chromatograph (GC) measurements, and the impact of re- duced sampling frequency on parameter uncertainty and flux estimates was evaluated. I would like to thank the authors for providing the data and code for this study. The paper is well written and is suitable for publication in Atmospheric Measure- ment Techniques. I have three general comments regarding the uncertainty analysis, the simulation of GC measurements, and the model assumptions.

*We would like to thank the reviewer for the thorough analysis and were very pleased to see that our data and code had been used. The comments are extremely valuable and we hope that our corresponding modifications to the manuscript will make it clearer and more understandable.*

**General comments**

RC 1. The analysis of the instrument stability and noise characteristics seems to suffer from repetition of several values that are very close to each other (see figure below). As a result, the Allan deviation analysis shows correlated noise up to 30 seconds averaging period. When the repeated values are removed, the Allan deviation shows a more realistic white noise behavior. These repeated values need to be removed before any further analysis. Note that they are not identical but they are very close to each other.

*AC 1. We are very grateful that the reviewer took the time to use our data and code and re-do some analyses. We agree with this first comment and thank the reviewer for pointing it out. With our FluxProGenie script, we found a way to remove and / or interpolate these repeated values with our filter script. Therefore, we will filter all data used for the analysis of the instrument stability and noise characteristics and re-do the analyses with the cleaned data.*

RC 2. The comparison with GC simulation is not rigorous enough in my opinion. Reporting underestimation from a down-sampled time series needs stronger justification. An increase in uncertainty from a fewer number of sampling points is expected and can be assessed from theory. That being said, a Monte Carlo simulation is also a valid approach to explore the increased uncertainty. However, in that case the subsampled time series should be taken from what is believed to be the true value of concentration (i.e. from the fitted model) and then noise can be added to it based on known GC uncertainty e.g. (Lebegue et al. 2016) (Rapson and Dacres 2014). Then the fluxes calculated from simulated GC measurements can be compared to PGA measurements. If the authors' argument is that there is more than just downsampling and that the concentration inside the camber fluctuates not only due to noise, then a justification needs to be provided for this argument. In that case taking mean downsampled PGA measurements is reasonable but needs explicit discussion of the uncertainty.

*AC 2. We agree that an increase in uncertainty from a time series with fewer data points can be expected and assessed from theory, and that the Monte Carlo simulation is a valid approach to explore uncertainties. However, from the comment, we understand that the reviewer may have come from a different approach than we did, namely a comparison between the different gas measurement methods vs. a comparison of the whole measurement procedure. If we assume that the only source of uncertainty is the precision of the method, namely the GC or the PGA, then a comparison of these methods with a Monte Carlo simulation would indeed be the suitable method to explore these uncertainties. However, our aim was to compare (previous) measurements analysed by a GC to those now conducted with portable gas analysers, assuming that, indeed, the concentration inside the chamber does fluctuate not only due to noise, because the fluxes are not stationary: the air in the chamber may not always be perfectly mixed and, with increasing chamber closure time, the soil underneath the chamber will get disturbed. As a result, we always see some scatter in the concentration data.*

*Because there are only very few samples when using the GC method, it is possible that the GC measuring points do not show normally distributed deviations from the actual trend line; this risk is significantly lower with the many PGA measuring points. We acknowledge that this was not discussed clearly enough. Therefore, we will restructure our result and discussion section, and add a few points to our discussion: "The underestimation of GC fluxes may occur as a result of a smoothed out curve: when only few data points are available, variations in curves are naturally reduced. Furthermore, the precision of our GC was 1.9 ppb compared to 0.2 ppb of the Aeris-$N_2O$, resulting in less accurate measurements of the $N_2O$ concentrations. This may lead to a loss of detail in the curve, particularly in the peak values of the $N_2O$ concentrations, which can result in underestimation of the flux. It is important to note that our comparison was made between our PGA and simulated GC measurements (Figure 7). For the GC simulations, we adjusted the instrument precision during the flux calculation, but no actual air samples were analysed by any GC instrument. Furthermore, our chamber closure time was considerably shorter than for most GC studies because of the condensation and temperature changes within the chamber. During prolonged chamber closure times, significant changes in the concentration gradient and chamber conditions can take place (see above), which are unlikely to be replicated in our GC-simulation. This difference in experimental design may actually be beneficial, as it allows us to isolate and study the effects of shorter closure times on $N_2O$ flux measurements. Furthermore, our sensitivity analysis with 4 simulated GC samples showed that even when we changed the sample times $\pm$ 60 sec compared to the original time stamp, flux rates differed less than 10%, with $R^2_{adj}$ values between 92 and 98 (data not shown). We believe that the underestimation of our $N_2O$ flux rates is, therefore, not a result of a inadequate simulation, but needs to be verified by future studies actually measuring $N_2O$ samples from nutrient-poor ecosystems in a GC."*

RC 3. It appears that both the LM and HM models assume that flux at the soil-air interface remains constant over time (time invariant). However, the study mentions that $N_2O$ fluxes change as the concentration gradient evolves within the chamber. Could the authors discuss this assumption more explicitly and its potential implications for flux calculations? Are there metrics that the authors recommend to assess the violations of this assumption?

*AC 3. We thank the reviewer for raising this important point. Indeed, both the LM and HM model assume that there is a*

*constant source concentration located at a certain depth within the soil (Hutchinson and Mosier, 1981). This assumption*

*is implied in the derivation of these models, who simulate a stable production rate of gases in the soil, resulting in a*

*gas accumulation within the chamber head space. The LM further assumes that the (diffusive) flux remains the same*

*throughout the chamber closure and that there are no chamber effects altering the diffusion rate, so that there is a stable,*

*linear increase or decrease of gases. This is where the HM model differs: it accounts for a non-linear curvature in the*

*chamber head space by allowing us to resolve the initial slope at time 0 (= in the least disturbed conditions); in other*

*words, solving for changes in concentration rates over time. This is based on non-linear changes in $N_2O$ concentrations*

*during closure time due to the reduction in the concentration gradient over time, as well as possible leakage (Hutchinson*

*and Mosier, 1981). Here, a citation from Kutzbach et al. 2007: "...for assessing the predeployment $CO_2$ flux, the rate*

*of initial concentration change at the moment of deployment (t=t0=0) should be used when the alteration of the con-*

*centration gradients in soils and plant tissues is minimal, rather than the mean rate of the $CO_2$ concentration change*

*over the chamber closure period (Livingston and Hutchinson, 1995). Many studies have investigated differences between*

*LM and HM models and addressed that the assumption of a stable flux may not hold due to, e.g., leakage, disturbance*

*of pressure gradients, or physical response of plants to temperature and moisture changes during the chamber closure*

*(Conen and Smith, 2000; Kutzbach et al., 2007; Maier et al., 2022; Creelman et al., 2013). The LM model is particu-*

*larly sensitive to this assumption about a stable flux rate because it fits a straight, linear line to the concentration-time*

*curve. As a result, fluxes calculated with the LM model can be seriously underestimated (Kutzbach et al., 2007). As*

*stated above, the HM model accounts for some non-linearity, but does not explicitly model a time-dependent soil flux,*

*and was shown to be more sensitive to random measurement errors (Venterea et al., 2020). This is why Hüppi et al.*

*(2018) introduced $\kappa$ as a decay constant for the concentration curve: if $\kappa$ is large, they assume a rapid deviation from*

*a constant-flux scenario, indicating that the assumption that the flux remains constant over time is weak. This is why*

*$\kappa_{max}$ is limited to the maximal curvature allowed in the model. Additionally, comparing the goodness-of-fit of different*

*models (e.g., Akaike Information Criterion corrected for small sample size (AICc)) can highlight when LM are not ad-*

*equate. With the best.flux function from the goFlux package, we can select the best flux estimates based on an objective*

*criteria, including $\kappa_{\max}$ and indices of model fit, namely MAE, RMSE, SE, and AICc (see here for more information:*

*https://qepanna.quarto.pub/goflux/bestflux.html). In that way, we believe that we account for the assumption and use the*

*available metrics to use the flux estimate closest to the "real" flux.*

**Specific comments**

RC 4. The title of the paper seems quite general. You might consider suggesting a more specific title that highlights the key contributions but I leave this to the authors to decide.

*AC 4. Thanks. We will change the title to "Practical guidelines for reproducible $N_2O$ flux chamber measurements in*

*nutrient-poor ecosystems".*

RC 5. 12 . . . "we can successfully detect" This sounds like you are reporting a detection limit. but you are reporting means with an uncertainty interval. I would suggest to rephrase this sentence.

*AC 5. We agree and will rephrase similar to "our results show that with our setup, we are able to detect and calculate*

*low $N_2O$ flux rates". Please see our abstract in the revised manuscript.*

RC 6. 15 MDF reported as a number with uncertainty interval is unusual. Maybe clarify what is reported here.

*AC 6. We agree that this can cause confusion. With the "goFlux" R package we use for flux calculation, MDF are given*

*for each measurement period (time the chamber remains closed) and calculated by dividing the precision of the PGA by*

*the exact time of the measurement period. Because we tested multiple chamber closure times, and measurement periods*

*may be up to 10 seconds shorter or longer depending on the filters used to correct the concentrations, this gives us a*

*range of values, which is represented in the numbers with uncertainty interval. For simplicity, we will, however, report*

*the MDF over all measurement periods in the abstract. Please see our abstract in the revised manuscript.*

RC 7. 20 "..or matched the linear model. .." could you clarify this sentence.

*AC 7. We agree and will rephrase either explain it better, e.g., "we used the non-linear model to calculate $N_2O$ fluxes, but*

*when the data were linearly distributed, the non-linear model produced the same results as a linear model, confirming*

*that the linear model was applicable in these cases.", or rephrase it similar to "we also found that the non-linear flux*

*calculation model yielded better results and was applicable in cases where the data were linearly distributed." Please*

*see our abstract in the revised manuscript.*

RC 8.   21 Underestimation flux when using GC needs stronger justification see my general comment.

*AC 8. Thanks; please see the response to the general comment 2.*

RC 9.   27 is the repetition of the IPCC full name necessary?

*AC 9. We will remove it in the revised manuscript.*

RC 10.   57 "At the same time, many of the reported $N_2O$ fluxes were found to be below the detection limit, making it challenging to assess the magnitudes and possible uptake of these fluxes." Not clear here which fluxes you are referring to. The phrase

"possible uptake"sounds ambiguous to me. do you mean "challenging to assess the sign of these fluxes"?

*AC 10. Thanks for the comment, we will rephrase to: "The detection limit was a significant constraint, as many reported*

$N_2O$ *fluxes were below the threshold of the GC method, limiting the ability to accurately assess their magnitude and*

*trends." We find the word trends more inclusive than the sign of these fluxes.*

RC 11.   91 the coordinates seem to be missing a dot "20.0'" instead of "200"

*AC 11. Thanks, we will correct that.*

RC 12.   180 where is the instrument precision taken from?

*AC 12. The instrument precision is provided by the manufacturer webpage which can be accessed through* [https://](https://)

[aerissensors.com/ultimate-precision-for-your-ghgs-measurementsthe-mira-co2-n2o/](https://aerissensors.com/ultimate-precision-for-your-ghgs-measurementsthe-mira-co2-n2o/) *.*

RC 13.   180 Definition of MDF is somehow strange as it does not include the number of points in the time series. Also this definition is different from what what is typically reported in the literature. For example (Maier et al. 2022) defines MDF

as ... I think it's also important to highlight that MDF defined here is only a theoretical limit based on the instrument uncertainty and does not account for model fit or chamber artifacts

*AC 13. We agree that the definition looks different from how it is typically reported in the literature. However, the equation itself is the same, although split into two equations (4) and (5)- only the water vapour correction is added. To avoid confusion, we will add the following: "Here, the MDF is a theoretical threshold that represents the instrument's detection limit, based on its precision ($\eta$) provided by the manufacturer. However, it does not account for potential errors in the model or chamber artefacts, but reflects the instrument's inherent uncertainty. The MDF can be calculated using Eq. 4 (...), where, $\theta$ is a flux term that corrects for the water vapour inside the chamber and converts the flux unit to $\mu mol$ $m^{-2}\ s^{-1}$ and t is the measurement time, i.e., the number of measurement points during the measurement period."*

RC 14. 194 "We used this factor because it has been previously used" I think better justification is needed here.

*AC 14. We agree and will change the statement to: "We used this factor because, upon visual assessment, it fit our data best, and has been previously used (Leiber-Sauheitl et al., 2014)."*

RC 15. 202 could you clarifies your second measurement strategy? did you take block averages or have you calculated the flux on 1-min intervals?

*AC 15. We suspect this question is due to unclear method explanations. In the field, chambers were closed for 10 min (= 600 data points and seconds (s)) for both light (transparent) and dark (opaque) measurements. For the assessment of different chamber closure times, we simply reduced the amount of data points by cutting seconds from the end of the measurement, i.e.: 540 s for 9 min, 480 s for 8 min, 420 s for 7 min, etc. We then calculated the fluxes with the remaining data points. To explore differences between light and dark measurements, we divided the resulting data set to be able to identify patterns and trends. We will rephrase to: "We first calculated all fluxes using the original 10-minute chamber closure time (prec = 0.2, g.limit = 4). To see how different closure times affect $N_2O$ fluxes, we shortened the closure time by 1 minute at a time, starting from 9 minutes, and recalculated the fluxes for each new time (e.g.,9 minutes = 540 seconds, 8 minutes = 480 seconds, etc.). We compared how chamber closure time affects flux rates in transparent and opaque measurements, and identified the number of fluxes above the minimum detectable flux based on the goFlux output."*

RC 16. 223 2500 ppm to 800 ppm seem very dry for ambient air, was their any drying involved?

*AC 16. Thanks for the comment. These were the conditions in laboratory (not the "outside" ambient air) whilst we did the 15 hrs- long sampling to investigate the instrument long-term drift and noise specifications. We conducted an experiment to examine the impact of water vapor using a dew-point generator; the results from that experiment were provided, please see Fig 4. We will adapt the text to: "From the 15-hour ambient air sampling in our closed laboratory, we observed that the water vapour mole fraction in the ambient air dropped from approximately 2500 ppm to about 800 ppm within the first 30 min."*

RC 17. Figure 1 description needs restructuring. Move information about shape files to the end if it is needed. Also shapefile is a technical term that might not be familiar to all readers. I would suggest to use a more general term like "geospatial data" or "spatial data" instead.

*AC 17. Thank you. We agree on the term shapefile being too technical and will use the suggested term "spatial data" instead.*

RC 18. Looking at Figure A1. I would not say there is no drift. At least in the beginning (maybe 2, 3 hours) there is a clear drift of about 1.5 ppb. You mentioned that 5 hours were needed for the signal to stabilize, would that mean the instrument needs 5 hours in the field for the signal to stabilize? or is that a temperature or pressure artifact? this needs more discussion.

*AC 18. Thank you. We agree that there is a drift, and Figure A1 is indeed meant to illustrate it. In the first five hours, we observed these fluctuations in the measurements partly because of the warm-up period then the signal stabilises. The implications of this were that we had to keep the instrument running continuously throughout the whole campaign, or make sure it was running for at least 5h before we used it in the field. The manufacturer are aware of this problem and recommended the approach we used in the field. Per reviewer's comment, we will add more discussion in the revised manuscript as follows: " Nevertheless, the long warm-up period (approximately 5 h) of the analyser needs to be taken into account, as this can be a limiting factor for certain applications. To mitigate this, the Aeris-$N_2O$ remained powered on throughout the whole field campaign"*

RC 19. Figure 4. is there a reason why relative humidity is chosen rather than absolute water concentration? For the analyzer, any matrix effects would be dependent on absolute water concentration rather than relative humidity.

*AC 19. Thank you. We agree that for the analyser, the absolute water concentration is the relevant variable. However, with this figure, we intended to show- in the most inclusive way- that with very different relative humidity (RH), the absolute $N_2O$ concentration of the analyser hardly changed. We consider that for most people, these high changes in RH are more intuitive and easier to understand than differing absolute water (vapour) concentrations. Therefore we would like to keep this as is.*

RC 20. 240 with such large uncertainty - this is basically not statistically different from zero - I would not conclude that the site acted as a sink. What is the number after the $\pm$ in the reported fluxes? Conventionally, it should be 2*SE to give a 95% confidence interval for the mean (can you please mention this when your first report it)

*AC 20. Thank you. We reported the standard deviation to show the spread of the data; however, we agree that that the standard error (SE) or the 95 % confidence interval are more relevant in this study. We will change our values accordingly and also indicate what we report. We argue that when giving SE or CI, the question of whether the site acted as a sink or not will not arise anymore. We will rephrase the first sentence to: "At our site, we commonly observed net $N_2O$ consumption, suggesting an atmospheric sink, with a mean flux of -0.469 $\mu g$ $N_2O$-N $m^{-2}h^{-1}$, with a 95% confidence interval (CI) of (-0.60,-0.3) during a chamber closure time of 10 min. Our calculated mean flux during transparent measurements was 0.361 $\mu g$ $N_2O$-N $m^{-2}h^{-1}$, with a 95% CI of (0.24,0.48) during a chamber closure time of 10 min (Table 1). For opaque measurements, our calculated flux was -1.29 $\mu g$ $N_2O$-N $m^{-2}h^{-1}$, with a 95% CI of (-1.45,-1.13), indicating that our opaque measurements represent a real biochemical process, rather than an experimental artefact, in the (sub-) Arctic ecosystem."*

RC 21. 245: is there something that supports this?

*AC 21. Thanks for the comment. We think that explaining the drivers of $N_2O$ fluxes, including environmental impacts and microbial processes, is out of scope of this study. The aim of this manuscript is to try answering the question of how to best capture low $N_2O$ fluxes in a nutrient-poor (sub-Arctic) ecosystem. We are currently working on analysing the impacts of environmental drivers on $N_2O$ fluxes; a follow up manuscript is in preparation. However, the reviewer is correct, as this statement needs to be supported. We will investigate the impact of $CO_2$ concentrations on the observed*

*N$_2$O fluxes under dark and light conditions, and are in touch with manufacturer and researchers who have been using the same instrument. According to their analysis, there is a crosstalk between CO$_2$ and N$_2$O signals (approximately -0..008 ppb of N$_2$O per a ppm of CO$_2$); however, this crosstalk would not change the outcome in our findings. Considering this, we will revise the statement as follows: "Nevertheless, the impact of environmental drivers on N$_2$O fluxes, including the transparent and opaque measurements, is beyond the scope of this study. Overall, we collected 338 samples, with 60-90 % of N$_2$O fluxes above the detectable limit. We therefore also acknowledge the possibility of unknown chamber artefacts that may remain undiscovered and could affect the interpretation of our data."*

RC 22. 269: I don't understand the recommendation for 3-5 minutes since the flux is continuously increasing. Could you elaborate more on this recommendation?

*AC 22. Thank you for raising this question. This suggestion was based on two main points: the minimal detectable flux (MDF) and soil disturbance when doing chamber measurements. As evident in Figure 5., the MDF and percentage of fluxes above the MDF change with respect to 10-min chamber closure time. Considering a 5 min chamber closure time, the difference between 10 and 5 min for dark measurements is around 15% and not statistically significant; the same holds for 3 and 4 min closure times. The second point is that N$_2$O availability through its diffusion into the soil is often the limiting factor for atmospheric N$_2$O consumption by N$_2$O reducing microbes. The N$_2$O diffusion to the soil is driven by the concentration gradient: when the chamber is closed, the N$_2$O concentration in the head space is decreasing as N$_2$O is diffusing into the soil. As a result, uptake rate is also decreasing, since N$_2$O reduction may become substrate limited. Consequently, long chamber closure times may underestimate the uptake of atmospheric N$_2$O and short chamber closure times should be favoured. We acknowledge that we did not specify this clearly enough and will restructure our manuscript as follows: "For opaque measurements, we find that our calculated fluxes show higher N$_2$O uptake from shorter chamber closure times, with flux rates around 15% lower at 3 - 5 min than at 10 min, respectively (Table 1). At 6 min, the differences in our calculated N$_2$O uptake was still 10% higher than at 10 min, decreasing to below 8% between 7 and 9 min. At the same time, the MDF increased from 56.5% to 87.1 between 3 min and 10 min (Fig. 5 b) ). Nevertheless, none of the flux rates across different closure times were significantly different from one another*

*(Kruskal-Wallis, p = 0.99). Especially for $N_2O$ uptake, it is essential to keep the chamber closure time as short as possible. This is because $N_2O$ availability through soil diffusion is often the limiting factor for microbial consumption, i.e., atmospheric $N_2O$ consumption by $N_2O$-reducing microbes (Liu et al., 2022). When the chamber is closed, the $N_2O$ concentration in the head space decreases as it diffuses into the soil, driven by the concentration gradient. As a result, the uptake rate also decreases, since $N_2O$ reduction may become substrate limited. Consequently, long chamber closure times may underestimate the uptake of atmospheric $N_2O$. Our analysis of the chamber closure time confirms this: during opaque measurement, we found that the uptake rate at 3-5 minutes were greatest, and decreased with every minute of added chamber closure time (Fig. 5). For opaque measurements, we therefore suggest to keep the chamber closure time between 3-5 min, unless very few data points are available, when aiming for fluxes above the MDF becomes more important."*

RC 23. 291: why does water increase in that pattern, and why would it decrease with time? Does temperature explain this behavior?

*AC 23. When chambers are closed onto the soil, we created a sealed environment, where atmospheric air should not be able to enter anymore. Because the initial headspace of the chamber is typically drier than the soil (especially in peatlands), water vapour from the soil, driven by the concentration gradient between the soil and the chamber, diffuses into the chamber: the H2O concentration within the chamber increases. This effect is particularly pronounced in the first 2 min after chamber closure. After some time, an equilibrium is reached, and the H2O concentration in the chamber remains relatively stable, as the rate of water vapour diffusion into the chamber and into the soil is equal. This can, indeed, be influenced by temperature: at higher temperatures, air can hold more water vapour, which can lead to a decrease in the absolute water vapour concentration. Since chambers act as a greenhouse, this effect gets more pronounced over time and was the reason why during the summer months, we hardly recorded any condensation inside the chamber.*

RC 24. Figure S5, is this a differenced time series? so is this ppm per minute or is it just normalized concentration by subtracting the first H2O measurement.

*AC 24. Thank you for pointing this out. The unit of the time interval incorrectly states minutes, when it should be seconds. We will correct this. What we want to illustrate here is the change in mean $H2O$ concentration (Figure S5) / temperature (Figure S6) inside the chamber across different time intervals. To do so, we created time intervals of 1 min, representing a 1-minute window of time with the aim to show more clearly that changes are much higher in the first minutes of the measurement period. We then summarise mean values for each minute and divide this into two plots for light and dark measurements. Hence, this figure is neither a differentiated time series, nor a plot of ppm per minute or a normalisation. It simply shows the mean values of $H2O$ / temperature inside the chamber across different time intervals (0 to 1 min, 1 to 2 min, 2 to 3 min, etc.) and light conditions (light and dark). We will add this to the figure caption.*

RC 25. 378 again I think an explanation is needed on why GC underestimates the flux

*AC 25. Thank you for this comment; we agree that this needs to be addressed more thoroughly. Please see our response to RC 2. for our overall answer.*

RC 26. in the conclusion, can you extend your recommendations to cover model choice, and how to avoid chamber artifacts. I am thinking of what metrics would you recommend to assess the goodness of fit of the model or when to detect the violations of the time-invariance assumption.

*AC 26. Thank you for this input. We agree that metrics to assess the goodness of fit of models or when / how to detect the violation of the time-invariance assumption are important. However, as stated in our response to RC 3., we aim to address this with the most novel flux calculation techniques available, namely the goFlux package, and think that recommendations of how to avoid chamber artefacts have been assessed by other studies, e.g., (Fiedler et al., 2022; Subke et al., 2021; Pumpanen et al., 2004)) and are out of scope for this study. Nevertheless, we will add some more references to that in the main text and modify our conclusion.*

**References**

Conen, F. and Smith, K. A.: An Explanation of Linear Increases in Gas Concentration under Closed Chambers Used to Measure Gas Exchange between Soil and the Atmosphere, European Journal of Soil Science, 51, 111–117, https://doi.org/10.1046/j.1365-2389.2000.00292.x, 2000.

Creelman, C., Nickerson, N., and Risk, D.: Quantifying Lateral Diffusion Error in Soil Carbon Dioxide Respiration Estimates Using Numerical Modeling, Soil Science Society of America Journal, 77, 699–708, https://doi.org/10.2136/sssaj2012.0352, 2013.

Fiedler, J., Fuß, R., Glatzel, S., Hagemann, U., Huth, V., Jordan, S., Jurasinski, G., Kutzbach, L., Maier, M., Schäfer, K., Weber, T., and Weymann, D.: BEST PRACTICE GUIDELINE: Measurement of carbon dioxide, methane and nitrous oxide fluxes between soil-vegetation-systems and the atmosphere using non-steady state chambers, 2022.

Hutchinson, G. L. and Mosier, A. R.: Improved Soil Cover Method for Field Measurement of Nitrous Oxide Fluxes, SOIL SCI. SOC. AM. J., 45, 6, 1981.

Hüppi, R., Felber, R., Krauss, M., Six, J., Leifeld, J., and Fuß, R.: Restricting the nonlinearity parameter in soil greenhouse gas flux calculation for more reliable flux estimates, PLOS ONE, 13, e0200 876, https://doi.org/10.1371/journal.pone.0200876, publisher: Public Library of Science (PLoS), 2018.

Kutzbach, L., Schneider, J., Sachs, T., Giebels, M., Nykänen, H., Shurpali, N. J., Martikainen, P. J., Alm, J., and Wilmking, M.: $CO_2$ flux determination by closed-chamber methods can be seriously biased by inappropriate application of linear regression, Biogeosciences, 4, 1005–1025, https://doi.org/10.5194/bg-4-1005-2007, 2007.

Leiber-Sauheitl, K., Fuß, R., Voigt, C., and Freibauer, A.: High $CO_2$ fluxes from grassland on histic Gleysol along soil carbon and drainage gradients, Biogeosciences, 11, 749–761, https://doi.org/10.5194/bg-11-749-2014, 2014.

Maier, M., Weber, T. K. D., Fiedler, J., Fuß, R., Glatzel, S., Huth, V., Jordan, S., Jurasinski, G., Kutzbach, L., Schäfer, K., Weymann, D., and Hagemann, U.: Introduction of a Guideline for Measurements of Greenhouse Gas Fluxes from Soils Using Non-steady-state Chambers, Journal of Plant Nutrition and Soil Science, 185, 447–461, https://doi.org/10.1002/jpln.202200199, 2022.

Pumpanen, J., Kolari, P., Ilvesniemi, H., Minkkinen, K., Vesala, T., Niinistö, S., Lohila, A., Larmola, T., Morero, M., Pihlatie, M., Janssens, I., Yuste, J. C., Grünzweig, J. M., Reth, S., Subke, J.-A., Savage, K., Kutsch, W., Østreng, G., Ziegler, W., Anthoni, P., Lindroth, A., and Hari, P.: Comparison of Different Chamber Techniques for Measuring Soil CO2 Efflux, Agricultural and Forest Meteorology, 123, 159–176, https://doi.org/10.1016/j.agrformet.2003.12.001, 2004.

Subke, J.-A., Kutzbach, L., and Risk, D.: Soil Chamber Measurements, in: Springer Handbook of Atmospheric Measurements, edited by Foken, T., pp. 1603–1624, Springer International Publishing, Cham, ISBN 978-3-030-52170-7 978-3-030-52171-4, https://doi.org/10.1007/978-3-030-52171-4_60, series Title: Springer Handbooks, 2021.

Venterea, R. T., Petersen, S. O., De Klein, C. A. M., Pedersen, A. R., Noble, A. D. L., Rees, R. M., Gamble, J. D., and Parkin, T. B.:

Global Research Alliance $N_2O$ Chamber Methodology Guidelines: Flux Calculations, Journal of Environmental Quality, 49, 1141–1155, https://doi.org/10.1002/jeq2.20118, 2020.